# DR-GGAD: Dual Residual Centering for Mitigating Anomaly Non-Discriminativity in Generalist Graph Anomaly Detection

**Changlong Fu,**[*] **Zhenli He,**[*] **Xiong Zhang, Cheng Xie,**[†] **Xin Jin, Yun Yang**
School of Software Engineering
Yunnan University, Kunming, China
`{hezl, xiecheng, xinjin, yangyun}@ynu.edu.cn`
`{fuchanglong, zhangxiong}@stu.ynu.edu.cn`

## Abstract

Generalist Graph Anomaly Detection (GGAD) seeks a unified representation learning model to detect anomalies in unseen graphs, but cross-domain transfer often entangles the learned anomalous and normal representations. We formalize this degradation as Anomaly non-Discriminativity ($\mathcal{A}n\mathcal{D}$) and define a normalized score to quantify it. We present DR-GGAD, which avoids direct comparison between anomalous and normal nodes via two residual modules: 1) a multi-scale Hyper Residual (HR) Center measuring node-to-center distances, yielding a compact normal residual structure with margin-pushed anomalies; 2) an Affinity-Residual (AR) module enforcing local residual directional consistency to recover structural separability. With frozen parameters (no target fine-tuning), DR-GGAD fuses both signals into a unified score. On 8 benchmark target graphs, it achieves new SOTA: mean AUROC +5.14% over the best prior GGAD, with large gains on high-$\mathcal{A}n\mathcal{D}$ datasets (ACM +9.96%, Amazon +7.48%) and strong AUPRC boosts (Amazon +17.12%, CiteSeer +17.77%). Ablations confirm complementary roles of the two modules. DR-GGAD thus establishes $\mathcal{A}n\mathcal{D}$ as a measurable bottleneck and delivers robust cross-domain anomaly detection.

## 1 Introduction

Learning robust representations on graphs is central to many mission-critical services. Effective representations allow social-media platforms to trace fake accounts and rumor cascades Duan et al. (2024); Yu et al. (2024), enable banks to uncover fraudulent transfers in massive ledgers Li et al. (2022a), and support security teams in blocking stealthy intrusions within corporate networks Jacob et al. (2022); Ghosh (2025). All of these tasks depend on Graph Anomaly Detection (GAD), whose goal is to pinpoint the few vertices whose attributes or links reveal malicious or abnormal behavior. Missing even a single anomaly can be costly; a high-frequency trading bot can distort prices in milliseconds, and a misinformation campaign can sway public sentiment before moderators react. Accuracy and timeliness are therefore essential.

**Limitations of domain-specific training.** Most GAD pipelines are trained on one graph at a time. They master the quirks of a single citation network, social platform, or e-commerce site and achieve strong in-domain accuracy. However, each new graph demands fresh labels, a hyper-parameter search, and additional compute. A security operations center that monitors many evolving networks soon faces prohibitive retraining costs, and models may become obsolete just as adversaries pivot to a new domain.

**The promise of generalization.** Generalist Graph Anomaly Detection (GGAD) aims to break this cycle by learning one detector that transfers to unseen graphs without further tuning. Recent work has taken encouraging steps: contextual residual learning in ARC Liu et al. (2024) and neighborhood

---

[*]These authors contributed equally.
[†]Corresponding author

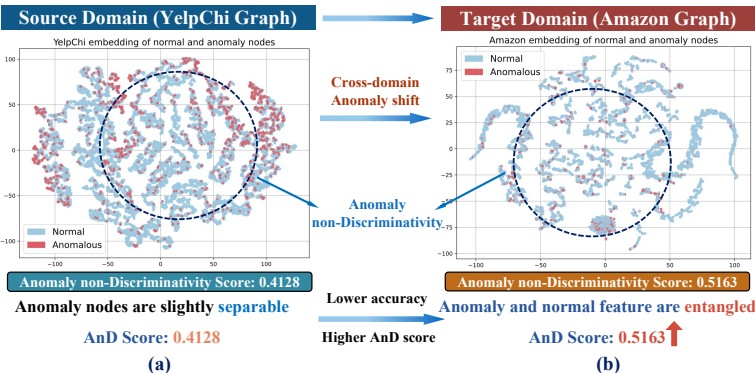

Figure 1: Anomaly non-Discriminativity ($\mathcal{A}n\mathcal{D}$). A GCN trained on YelpChi transfers to Amazon; the $\mathcal{A}n\mathcal{D}$ score rises and AUC falls.

prompting in UNPrompt Niu et al. (2025) both improve zero-shot performance. A critical obstacle, however, continues to limit progress.

**The Anomaly non-Discriminativity ($\mathcal{A}n\mathcal{D}$) Bottleneck.** $\mathcal{A}n\mathcal{D}$ measures how much the latent representations of normal and anomalous nodes overlap; higher values indicate weaker separability. Figure 1(a) shows a Graph Convolutional Network (GCN) trained on the YelpChi review graph. Its embeddings are already partly entangled, yielding an $\mathcal{A}n\mathcal{D}$ score of 0.41. After transfer to Amazon's larger co-review graph, differences in feature statistics and neighborhood patterns blur the decision boundary. The $\mathcal{A}n\mathcal{D}$ score climbs to 0.52, and the area under the ROC curve (AUC) falls from 0.58 to 0.46 [Figure 1(b)].

**Practical consequences.** Imagine the opening morning of a holiday sale. Trust-and-safety engineers rely on this detector to intercept coordinated fake-review rings. Thousands of five-star posts arrive within hours. Because the elevated $\mathcal{A}n\mathcal{D}$ score hides abnormal users among legitimate shoppers, roughly one in five fraudulent accounts slips through automated screening until a manual audit. By that point, product rankings are distorted, and purchasing decisions affecting millions of dollars have already been influenced.

**Our solution: compare with a center rather than with each other.** Direct comparison between anomalies and normals is fragile under domain shift. We instead measure how both groups deviate from two residual centers. Concretely, it comprises two components: **(1) Hyper Residual (HR) Center.** Multi-scale residuals from successive graph neural network layers form a domain-invariant reference. Normal nodes cluster tightly around this center, while anomalies are pushed outward, restoring feature-space separability. **(2) Affinity Residual (AR) Center.** Residual vectors are aligned within structural neighborhoods, revealing anomalies that hide primarily in topology rather than attributes. With parameters frozen after source training, **DR-GGAD** fuses HR and AR signals into a single score that transfers to target graphs without tuning.

**Impact in practice.** Across eight public benchmarks, DR-GGAD raises mean AUC by **5.14%** over the previous GGAD best. Gains are especially large on high-$\mathcal{A}n\mathcal{D}$ graphs: ACM (+9.96%), and Amazon (+7.48%). The area under the precision–recall curve (AUPRC) improves in parallel, climbing 17.12% on Amazon and 17.77% on CiteSeer. These gains correspond to many more fraudulent users and compromised hosts being caught before damage occurs.

**Contributions of this work.** (1) We isolate and quantify $\mathcal{A}n\mathcal{D}$ and release a normalized $\mathcal{A}n\mathcal{D}$ score for reproducible benchmarking. (2) We introduce *Dual Residual Centering*: the HR center mitigates feature-space entanglement, and the AR center restores structural alignment, yielding a detector that needs no target-domain tuning. (3) We conduct extensive experiments that establish new state-of-the-art results on eight benchmarks, demonstrating both the importance of addressing $\mathcal{A}n\mathcal{D}$ and the effectiveness of DR-GGAD. These advances move GGAD closer to deployment-ready reliability, allowing practitioners to monitor emerging graphs without continual retraining.

## 2 RELATED WORK

### 2.1 GRAPH ANOMALY DETECTION

Early GAD research framed node-level detection as a reconstruction or similarity task. Autoencoder variants rebuild graph structure and attributes to highlight large reconstruction errors as anomalies Ding et al. (2019); Zheng et al. (2023); He et al. (2024). Deep SVDD Ruff et al. (2018) and Deep SAD Ruff et al. (2020) perform anomaly detection by compressing the representations of normal samples into a hypersphere, making them suitable for single-domain tasks. However, their performance may be affected when facing domain shifts, limiting their application in cross-domain tasks. Distance-based methods compute local or global affinity scores; examples include OCGNN Wang et al. (2021) and the reinforcement-guided strategy in Bei et al. (2023). Edge-truncation models such as TAM Qiao & Pang (2023) and GCTAM Zhang et al. (2025) prune heterophilous edges, improving normal-node homophily. Contrastive learning further sharpens decision boundaries by sampling positive and negative node pairs Liu et al. (2021); Kim et al. (2023). Spectral and propagation-aware designs—GHRN Gao et al. (2023), BWGNN Tang et al. (2022), and SmoothGNN Dong et al. (2025)—trace the flow of high-frequency signals to flag outliers. Recent adversarial and semi-supervised methods generate pseudo-anomalies or perturbations, raising robustness and data efficiency Ding et al. (2021a); Meng et al. (2023); Qiao et al. (2024); Li et al. (2022b); Xiao et al. (2023).

**Limitations.** Most existing methods assume similar train–test distributions, so their representations degrade on unseen domains—a failure captured by our $\mathcal{AnD}$ metric. They further rely on labeled anomalies or domain-specific tuning, conflicting with the zero-tuning goal of GGAD.

### 2.2 GENERALIST GRAPH ANOMALY DETECTION

GGAD seeks a single detector that generalizes across graphs. Previous cross-domain methods like COMMANDER Ding et al. (2021b), ACT Wang et al. (2023), and CDFS-GAD Chen et al. (2024) align node representations or anomaly patterns to handle cross-domain tasks effectively. However, they rely on target-domain data or fine-tuning, limiting their applicability in cases of data scarcity or unknown target domains. Recent works, such as ARC Liu et al. (2024) sorts node features by smoothness and employs a multi-hop residual encoder to extract transferable signals under few-shot settings. UNPrompt Niu et al. (2025) introduces unified neighborhood prompts that guide a zero-shot model without any fine-tuning. AnomalyGFM Qiao et al. (2025) pre-trains graph-agnostic prototypes for normal and abnormal classes, enabling both zero-shot and few-shot use.

**Our distinction.** Although these methods improve transferability, none explicitly tackle the representation overlap we term $\mathcal{AnD}$. DR-GGAD mitigates this bottleneck through Dual Residual Centering: the Hyper Residual center reduces feature-space entanglement, and the Affinity Residual center restores structural separability, together delivering a frozen model that remains effective across a wide range of unseen graphs.

| Method | Cross-Domain | Addresses $\mathcal{AnD}$ | Needs Few Samples | Key Idea |
|---|---|---|---|---|
| TAM Qiao & Pang (2023) | No | No | Yes | Edge truncation |
| GHRN Gao et al. (2023) | No | No | Yes | High-freq signals |
| UNPrompt Niu et al. (2025) | Yes | No | No | Neighborhood prompt |
| ARC Liu et al. (2024) | Yes | No | Yes | Contextual residual |
| AnomalyGFM Qiao et al. (2025) | Yes | No | No | Prototype matching |
| **DR-GGAD (Ours)** | Yes | Yes | No | Dual residual centers |

Table 1: Comparison with representative GAD and GGAD methods.

**Innovation Highlight.** Table 1 contrasts DR-GGAD with prior art. Our method is the first to (1) quantify $\mathcal{AnD}$, (2) address it jointly in feature and structure space, and (3) achieve state-of-the-art transfer without any target-domain tuning.

## 3 PROBLEM FORMULATION

### 3.1 NOTATION AND PRELIMINARIES

Let $\mathcal{G} = (\mathcal{V}, \mathbf{A}, \mathbf{X})$ be an *attributed graph* with node set $\mathcal{V} = \{v_1, v_2, \ldots, v_N\}$, adjacency matrix $\mathbf{A} \in \{0,1\}^{N \times N}$ where $\mathbf{A}_{ij} = 1$ if and only if an undirected edge connects $v_i$ and $v_j$, and feature matrix $\mathbf{X} \in \mathbb{R}^{N \times d}$ whose $i$-th row $\mathbf{x}_i$ is the $d$-dimensional attribute of $v_i$. A frozen encoder $H : \mathbb{R}^d \to \mathbb{R}^h$ maps each $\mathbf{x}_i$ to an embedding $\mathbf{z}_i = H(\mathbf{x}_i)$. We assume $|N^+| \geq 1$ and $|N^-| \geq 1$ whenever the quantities below are defined.

### 3.2 GENERALIST GRAPH ANOMALY DETECTION TASK

We are given $n_s$ labeled *source* graphs $\mathcal{T}_{\text{train}} = \{\mathcal{D}_{\text{train}}^{(1)}, \ldots, \mathcal{D}_{\text{train}}^{(n_s)}\}$, each $\mathcal{D}_{\text{train}}^{(k)} = (\mathcal{G}^{(k)}, \mathbf{y}^{(k)})$ with node labels $\mathbf{y}^{(k)} \in \{0,1\}^{|\mathcal{V}^{(k)}|}$ ($1 =$ anomaly). A detector $f_\theta$ is learned on all sources and then **frozen**. At deployment, $f_\theta$ is applied *without fine-tuning* to every unseen *target* graph in $\mathcal{T}_{\text{test}} = \{\mathcal{G}_{\text{test}}^{(1)}, \ldots, \mathcal{G}_{\text{test}}^{(n_t)}\}$. For each target node $v_i$, the goal is to output a score $s_i \in \mathbb{R}$ that reflects its anomaly likelihood.

### 3.3 ANOMALY NON-DISCRIMINATIVITY ($\mathcal{A}n\mathcal{D}$)

**Unnormalized score.** Let $N^+$ and $N^-$ denote the sets of normal and anomalous nodes. Their average pairwise Euclidean distances are

$$d^{(+)} = \frac{1}{|N^+|^2} \sum_{i \in N^+} \sum_{j \in N^+} \|\mathbf{z}_i - \mathbf{z}_j\|_2, \ d^{(-)} = \frac{1}{|N^-|^2} \sum_{i \in N^-} \sum_{j \in N^-} \|\mathbf{z}_i - \mathbf{z}_j\|_2,$$
$$d^{(+-)} = \frac{1}{|N^+| |N^-|} \sum_{i \in N^+} \sum_{j \in N^-} \|\mathbf{z}_i - \mathbf{z}_j\|_2. \tag{1}$$

Then, we define

$$\mathcal{A}n\mathcal{D}^*(\mathcal{G}) = d^{(+)} + d^{(-)} - d^{(+-)}. \tag{2}$$

*Note.* We use V-statistics; the corresponding U-statistic versions are asymptotically equivalent for our purposes.

**Normalization.** Fix an evaluation suite $\mathbb{S}$ of graphs used for cross-dataset comparison. To compare across graphs, we linearly rescale over $\mathbb{S}$:

$$\mathcal{A}n\mathcal{D}(\mathcal{G}) = \frac{\mathcal{A}n\mathcal{D}^*(\mathcal{G}) - \min_{\mathcal{G}' \in \mathbb{S}} \mathcal{A}n\mathcal{D}^*(\mathcal{G}')}{\max_{\mathcal{G}' \in \mathbb{S}} \mathcal{A}n\mathcal{D}^*(\mathcal{G}') - \min_{\mathcal{G}' \in \mathbb{S}} \mathcal{A}n\mathcal{D}^*(\mathcal{G}')}, \tag{3}$$

where $\mathcal{A}n\mathcal{D}(\mathcal{G}) \in [0,1]$ is the normalized $\mathcal{A}n\mathcal{D}$ score. A higher $\mathcal{A}n\mathcal{D}(\mathcal{G})$ score indicates greater overlap and weaker separability, while a lower score reflects clearer separation and stronger discriminability.

### 3.4 THEORETICAL PROPERTIES OF $\mathcal{A}n\mathcal{D}$

We record two minimal properties to justify $\mathcal{A}n\mathcal{D}$ as a separability surrogate before presenting DR-GGAD.

**Proposition 1** (Range & Calibration). *Let $m = \min_{\mathbb{S}} \mathcal{A}n\mathcal{D}^*$ and $M = \max_{\mathbb{S}} \mathcal{A}n\mathcal{D}^*$ with $M > m$. Then for any $\mathcal{G} \in \mathbb{S}$, $\mathcal{A}n\mathcal{D}(\mathcal{G}) = ((\mathcal{A}n\mathcal{D}^*(\mathcal{G}) - m)/(M - m)) \in [0,1]$, with $\mathcal{A}n\mathcal{D}(\mathcal{G}) = 0 \iff \mathcal{A}n\mathcal{D}^*(\mathcal{G}) = m$ and $\mathcal{A}n\mathcal{D}(\mathcal{G}) = 1 \iff \mathcal{A}n\mathcal{D}^*(\mathcal{G}) = M$; if $M = m$, set $\mathcal{A}n\mathcal{D} \equiv 0$.*

**Lemma 2** (Lipschitz scoring gap). *For any $L$-Lipschitz score $g : \mathbb{R}^h \to \mathbb{R}$ applied to embeddings, if $(i, j)$ is drawn uniformly from $N^+ \times N^-$, then*

$$\mathbb{E}_{(i,j) \sim \text{Unif}(N^+ \times N^-)} \big[ |g(\mathbf{z}_j) - g(\mathbf{z}_i)| \big] \leq L\, d^{(+-)}. \tag{4}$$

*Proof.* By Lipschitz continuity, $|g(\mathbf{z}_j) - g(\mathbf{z}_i)| \leq L\|\mathbf{z}_j - \mathbf{z}_i\|_2$ for any $(i, j) \in N^+ \times N^-$. Taking expectations over $(i, j) \sim \text{Unif}(N^+ \times N^-)$ yields the claim. $\square$

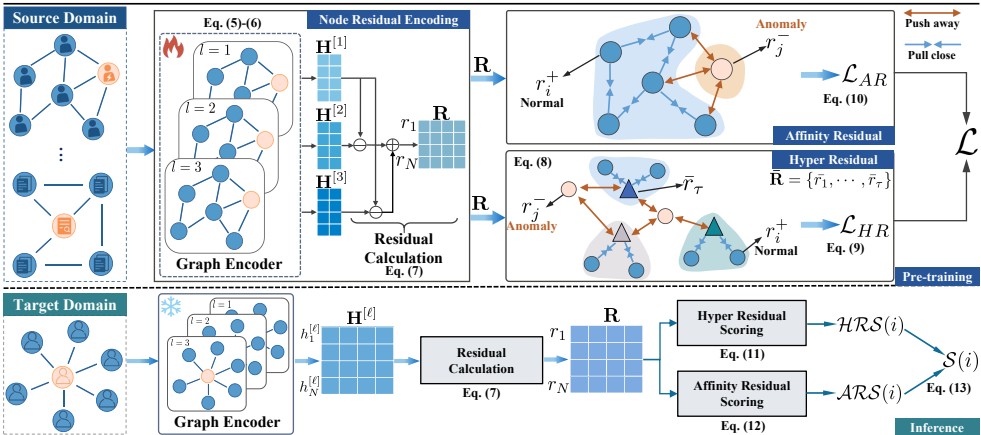

Figure 2: Overview of DR-GGAD.

*Remark.* When the class-conditional embedding distributions coincide, $\mathcal{A}n\mathcal{D}^*$ typically lies near the upper end on a given suite $\mathbb{S}$; conversely, well-separated embeddings tend to the lower end. Whether such extremes are realized depends on $\mathbb{S}$.

## 3.5 KEY CHALLENGES

**C1: Distribution shift.** Feature statistics, degree profiles, and homophily ratios vary widely across graphs, eroding the margin learned in the source domain. **C2: No adaptation budget.** Target graphs arrive without labels, and the detector is not allowed to update its parameters. **C3: Embedding overlap.** When C1 and C2 coincide, normal and anomalous embeddings overlap, which is reflected by a high $\mathcal{A}n\mathcal{D}$ score.

**Design Objective.** DR-GGAD combats C3 by introducing two residual *centers* that provide a stable, graph-agnostic reference in both feature and structure space, thereby counteracting C1 while respecting the no-adaptation constraint C2. In DR-GGAD, HR compresses $d^{(+)} + d^{(-)}$ while AR enlarges $d^{(+-)}$; by equation 2, these jointly decrease $\mathcal{A}n\mathcal{D}^*$. Collectively, the above properties support $\mathcal{A}n\mathcal{D}$ as a useful surrogate for separability: lower values coincide with reduced class overlap and, empirically, correlate with stronger cross-domain generalization. To further investigate the $\mathcal{A}n\mathcal{D}$ challenge, we conduct a quantitative analysis of both $\mathcal{A}n\mathcal{D}^*$ and normalized $\mathcal{A}n\mathcal{D}$ scores, with detailed results provided in Appendix A.

## 4 METHODOLOGY

To address Anomaly non-Discriminativity ($\mathcal{A}n\mathcal{D}$) from direct node comparisons, DR-GGAD shifts to analyzing self-residuals. We propose dual residual centering with Hyper Residual and Affinity Residual, anchoring each node to two domain-invariant centers and replacing fragile normal-to-abnormal contrast with more stable self-to-self discriminability. This section details how we (1) extract multi-layer residual embeddings, (2) build the Hyper Residual (HR) center, (3) enforce local Affinity Residual (AR) coherence, and (4) fuse both signals into a single anomaly score. An overview is provided in Figure 2.

### 4.1 NODE RESIDUAL ENCODING

The encoder transforms raw features into residual embeddings that capture cross-layer variations—signals known to transfer well across graphs.

**Node attribute alignment.** Heterogeneous input dimensions are unified by a linear or random projection, i.e., $\bar{\mathbf{X}} = \mathcal{T}(\mathbf{X}) \in \mathbb{R}^{N \times d_u}$, where $\mathcal{T}(\cdot)$ preserves discriminative information while enforcing a unified dimension $d_u$.

**Shared graph encoder.** Aligned features propagate through $\ell$ light graph convolution layers with shared parameters:

$$\tilde{\mathbf{X}}^{[1]} = \mathbf{D}^{-\frac{1}{2}}\mathbf{A}\mathbf{D}^{-\frac{1}{2}}\bar{\mathbf{X}}, \ldots, \tilde{\mathbf{X}}^{[\ell]} = \mathbf{D}^{-\frac{1}{2}}\mathbf{A}\mathbf{D}^{-\frac{1}{2}}\tilde{\mathbf{X}}^{[\ell-1]}, \tag{5}$$

followed by projections

$$\mathbf{H}^{[t]} = \sigma\big(\tilde{\mathbf{X}}^{[t]}\mathbf{W}^{[t]}\big) \in \mathbb{R}^{N \times d_h}, \quad t = 1, \ldots, \ell. \tag{6}$$

**Residual calculation.** Layer-wise differences highlight how each node reacts to wider receptive fields:

$$r_i^{[t]} = h_i^{[t]} - h_i^{[1]}, \ 1 < t \leq \ell, \qquad r_i = r_i^{[2]} \| r_i^{[3]} \| \cdots \| r_i^{[\ell]}. \tag{7}$$

Stacking all $r_i$ yields $\mathbf{R} \in \mathbb{R}^{N \times (\ell-1)d_h}$. $r_i$ not only reveals the differences between the ego node and its neighbors but also effectively captures high-frequency signals and local heterophily. This enables the model to capture consistent residual representations shared across domains, which is crucial for anomaly detection Qiao & Pang (2023), Liu et al. (2024). At the same time, it preserves the ability to identify more complex anomaly patterns within the graph structure.

### 4.2 HYPER RESIDUAL LEARNING

Instead of contrasting normal and abnormal residuals directly, we summarize normal patterns into a compact set of centroids and measure each node against them.

**Center construction.** Across all source graphs, $k$-means clusters normal residuals into $\tau$ groups. Each cluster $\mathbf{C}_i$ contains normal residuals assigned to the $i$-th group. At iteration $t$, the centroids update as

$$\bar{r}_i^{(t+1)} = \sigma\Big(\frac{\mathbf{W}_i}{|\mathbf{C}_i^{(t)}|} \sum_{r_j^{+(t)} \in \mathbf{C}_i^{(t)}} r_j^{+(t)}\Big), \mathbf{C}^{(t)} = \texttt{Clustering}(\mathbf{R}^{+(t-1)}, \tau), \ \mathbf{C} \in \mathbb{R}^{\tau \times (\ell-1)d_u}, \tag{8}$$

forming the global HR center set $\bar{\mathbf{R}} = \{\bar{r}_1, \ldots, \bar{r}_\tau\}$, and the centers are constructed based on the normal node residuals $r_j^+$, reflecting the typical patterns of normal nodes.

**Center-based loss.** Normal residuals are drawn toward the centers, while anomalies are repelled:

$$\mathcal{L}_{HR} = \sum_{i=1}^{N^+}\sum_{k=1}^{\tau} \big\|r_i^+ - \bar{r}_k\big\|_2 + \sum_{i=1}^{N^-}\sum_{k=1}^{\tau} \max\big(0, \epsilon - \big\|r_i^- - \bar{r}_k\big\|_2\big), \tag{9}$$

where $r_i^+$ and $r_i^-$ denote the residuals of the normal and anomalous node, respectively. By minimizing $\mathcal{L}_{HR}$, we encourage the residuals of normal nodes to be close to the HR center, thereby enhancing their semantic consistency, while forcing the residuals of anomalous nodes to exceed the boundary $\epsilon$, ensuring a clear separation between normal and anomalous nodes. This ensures a clear separation between normal and anomalous nodes, consequently amplifying the anomalous semantic information of anomalous nodes in the graph.

### 4.3 AFFINITY RESIDUAL LEARNING

While the HR center enforces global alignment, local structures may still diverge. AR learning, therefore, aligns residual directions within each neighborhood.

**Affinity Score and Consistency Loss.** To promote structural consistency in residual space, we define the affinity-based consistency loss as:

$$\mathcal{L}_{AR} = \sum_{i=1}^{N}\big(1 - \mathcal{AR}(i)\big), \quad \mathcal{AR}(i) = \frac{1}{|N_i|}\sum_{j \in N_i}\frac{r_i \cdot r_j}{\|r_i\|\,\|r_j\|}, \tag{10}$$

where $\mathcal{AR}(i)$ measures the average cosine similarity between node $i$ and its neighbors in the residual representation space. Minimizing $\mathcal{L}_{AR}$ strengthens local structural semantics by ensuring that neighboring nodes with regular structural patterns remain aligned in the residual space, while nodes

with structural anomalies or semantic inconsistencies exhibit deviations. By imposing this constraint, it helps model capture meaningful local structural features, enabling it to detect anomalies without relying on reconstruction-based supervision, thus alleviating the structural entanglement underlying the $\mathcal{A}n\mathcal{D}$ problem.

**Joint objective.** Both losses are combined to learn transferable residual representations and are defined as $\mathcal{L} = \mathcal{L}_{HR} + \mathcal{L}_{AR}$.

## 4.4 ANOMALY SCORING AND PREDICTION

After training on sources, all parameters are frozen. For each target graph, we compute two complementary scores.

**Hyper Residual score (HRS).** Feature-level deviation from the HR centers is

$$\mathcal{HRS}(i) = \frac{1}{\tau} \sum_{k=1}^{\tau} \|r_i - \bar{r}_k\|_2, \tag{11}$$

where larger values indicate stronger abnormality in residual space.

**Affinity Residual score (ARS).** We quantify how much node $i$ deviates from its neighborhood:

$$\mathcal{ARS}(i) = \frac{1}{|N_i|} \sum_{j \in N_i} \left(1 - \frac{r_i \cdot r_j}{\|r_i\| \, \|r_j\|} + \|r_i - r_j\|_2\right). \tag{12}$$

**Unified score.** The final anomaly score aggregates both perspectives:

$$\mathcal{S}(i) = \lambda \, \mathcal{HRS}(i) + (1 - \lambda) \, \mathcal{ARS}(i), \quad \lambda \in [0, 1]. \tag{13}$$

**Threshold selection.** We choose the threshold that maximizes separation between predicted normals and anomalies:

$$\begin{cases} \gamma^* = \arg \max_{\gamma \in \{\mathcal{S}(i)|i \in N\}} \left[\frac{1}{|N^+|} \sum_{i \in N^+} \mathbb{I}(\mathcal{S}(i) \geq \gamma) - \frac{1}{|N^-|} \sum_{j \in N^-} \mathbb{I}(\mathcal{S}(j) \geq \gamma)\right], \\ \hat{y}_i = \begin{cases} 1, & \mathcal{S}(i) \geq \gamma^*, \\ 0, & \mathcal{S}(i) < \gamma^*. \end{cases} \end{cases} \tag{14}$$

This dual-center strategy directly targets anomaly non-discriminativity by shrinking residual overlap in both feature and structure spaces, enabling robust zero-shot detection on unseen graphs.

## 5 EXPERIMENTS

We now demonstrate that DR-GGAD converts the theoretical gains promised by dual residual centering into tangible improvements on eight heterogeneous target graphs.

### 5.1 EXPERIMENTAL SETUP

**Datasets.** We follow ARC's cross-domain protocol Liu et al. (2024). The four *source* graphs are $\mathcal{T}_{\text{train}}$ = {PubMed, Flickr, Questions, YelpChi}, and the eight *target* graphs are $\mathcal{T}_{\text{test}}$ = {Cora, CiteSeer, ACM, BlogCatalog, Facebook, Weibo, Reddit, Amazon}. Each target mixes synthetically injected and naturally occurring anomalies, ensuring a rigorous evaluation of cross-graph generalization.

**Baselines.** We compare DR-GGAD against sixteen strong methods: (1) *GAD methods and Supervised Pre-Train*—GCN Kipf & Welling (2016), GAT Veličković et al. (2017), DOMINANT Ding et al. (2019), BGNN Ivanov & Prokhorenkova (2021), BWGNN Tang et al. (2022), GHRN Gao et al. (2023); *GAD methods and Unsupervised Pre-Train*—CoLA Liu et al. (2021), HCM-A Huang et al. (2022), TAM Qiao & Pang (2023), SmoothGNN Dong et al. (2025), GCTAM Zhang et al. (2025), CAGAD Xiao et al. (2024), and Semi-GGAD Qiao et al. (2024); (2) *GGAD methods and*

*Supervised Pre-Train*—UNPrompt Niu et al. (2025), AnomalyGFM Qiao et al. (2025), and ARC Liu et al. (2024).

**Metrics and implementation.** AUROC and AUPRC are reported with mean $\pm$ std over five seeds Tang et al. (2023). All runs use an RTX A6000 GPU and a 12th-Gen Intel i7-12700 CPU. Unless noted, the learning rate is $10^{-5}$, weight decay $5 \times 10^{-5}$, dropout 0.2, unified feature dimension $d_u = 64$, residual dimension $d = 1024$, GCN layers $\ell = 3$, separation margin $\epsilon = 1$, and 60 training epochs.

Table 2: **Anomaly detection performance (AUROC, %, mean $\pm$ std).** "Rank" is the average rank over eight targets; $\Delta$ is the absolute AUROC gain over the second-best method. The best, second, and third results are colored first, second, and third, respectively. Results tagged "*" are reproduced with the authors' code and default configures; all others are copied from ARC's Liu et al. (2024) leader-board.

| Method | Facebook | ACM | Amazon | Cora | CiteSeer | Reddit | Weibo | BlogCatalog | Rank |
|---|---|---|---|---|---|---|---|---|---|
| GAD Methods & Supervised - Pre-Train | | | | | | | | | |
| GCN(2017) | 29.51±4.86 | 60.49±9.65 | 46.63±3.47 | 59.64±8.30 | 60.27±8.11 | 50.43±4.41 | 76.64±17.69 | 56.19±6.39 | 10.00 |
| GAT(2018) | 51.88±2.16 | 48.79±2.73 | 50.52±17.22 | 50.06±2.65 | 51.59±3.49 | 51.78±4.04 | 53.06±7.48 | 50.40±2.80 | 10.75 |
| DOMINANT(2019) | 51.01±0.78 | 70.08±2.34 | 48.94±2.69 | 66.53±1.15 | 69.47±2.02 | 50.05±4.92 | 92.88±0.32 | 74.25±0.65 | 6.88 |
| BGNN(2021) | 54.74±25.29 | 44.00±13.69 | 52.26±3.31 | 42.45±11.57 | 42.32±11.82 | 50.27±3.84 | 32.75±35.35 | 47.67±8.52 | 13.62 |
| BWGNN(2022) | 45.84±4.97 | 67.59±0.70 | 55.26±16.95 | 54.06±3.27 | 52.61±2.88 | 48.97±5.74 | 53.38±1.61 | 56.34±1.21 | 10.31 |
| GHRN(2023) | 44.81±8.06 | 55.65±6.37 | 49.48±17.13 | 59.89±6.57 | 56.04±9.19 | 46.22±2.33 | 51.87±14.18 | 57.64±3.48 | 11.00 |
| GAD Method & Unsupervised - Pre-Train | | | | | | | | | |
| CoLA(2021) | 12.99±11.68 | 66.85±4.43 | 47.40±7.97 | 63.29±8.88 | 62.84±9.52 | 52.81±6.69 | 16.27±5.64 | 50.04±3.25 | 11.00 |
| HCM-A(2022) | 35.44±13.97 | 53.70±4.64 | 43.99±0.72 | 54.28±4.73 | 48.12±6.80 | 48.79±2.75 | 65.52±12.58 | 55.31±0.57 | 12.88 |
| TAM(2023) | 65.88±6.66 | 74.43±1.59 | 56.06±2.19 | 62.02±2.39 | 72.27±0.83 | 55.43±0.33 | 71.54±0.18 | 49.86±0.73 | 5.62 |
| CAGAD(2024)* | 45.84±4.97 | 39.80±9.91 | 46.06±0.75 | 50.11±3.41 | 40.13±5.41 | 54.57±3.89 | 58.99±3.42 | 49.84±12.37 | 13.44 |
| Semi-GGAD(2024)* | 55.89±8.99 | 37.47±2.68 | 53.11±4.92 | 39.44±5.41 | 38.18±4.21 | 55.39±0.44 | 65.73±3.35 | 50.70±7.34 | 11.38 |
| SmoothGNN(2025)* | 48.81±6.05 | 49.02±7.37 | 51.77±26.49 | 51.52±5.39 | 49.61±5.80 | 51.13±2.76 | 37.14±21.32 | 49.51±5.86 | 12.75 |
| GCTAM(2025)* | 69.57±1.41 | 81.21±0.13 | 55.74±0.60 | 58.78±2.17 | 70.31±1.77 | 59.32±0.73 | 70.61±0.10 | 67.60±0.77 | 4.62 |
| GGAD Methods & Supervised - Pre-Train | | | | | | | | | |
| UNPrompt(2025)* | 55.27±6.90 | 69.91±1.28 | 56.02±11.69 | 54.31±1.50 | 49.80±3.12 | 59.18±1.44 | 45.56±3.75 | 68.36±0.40 | 7.75 |
| AnomalyGFM(2025)* | 58.64±7.14 | 60.79±1.48 | 60.65±9.07 | 54.17±3.08 | 54.71±2.30 | 59.99±1.69 | 69.48±11.11 | 57.77±3.31 | 6.62 |
| ARC(2024) | 67.56±1.60 | 79.88±0.28 | 80.67±1.81 | 87.45±0.74 | 90.95±0.59 | 60.04±0.69 | 88.85±0.14 | 74.76±0.06 | 2.38 |
| Our & Supervised - Pre-Train | | | | | | | | | |
| DR-GGAD | 82.16±1.98 | 91.17±0.78 | 88.15±2.21 | 93.20±0.56 | 95.00±0.41 | 60.60±0.62 | 93.31±0.22 | 75.06±0.09 | 1.00 |
| $\Delta$ | 12.59↑ | 9.96↑ | 7.48↑ | 5.75↑ | 4.05↑ | 0.56↑ | 0.43↑ | 0.30↑ | - |

## 5.2 OVERALL PERFORMANCE

As summarized in Table 2, DR-GGAD posts the top AUROC on every one of the eight target graphs, yielding a perfect average rank of 1.0. Its largest margins—Facebook (+12.59%), ACM (+9.96%), and Amazon (+7.48%)—coincide with the datasets that exhibit the greatest $\mathcal{AnD}$. Traditional supervised detectors (GCN, GAT, BGNN, GHRN) collapse once the deployment graph drifts away from the training distribution; unsupervised methods (e.g., DOMINANT, GCTAM) thrive under strong homophily but falter on heterogeneous structures. Earlier GGAD designs, such as UNPrompt and ARC transfer better, yet they still compare nodes directly and thus lose separation when residual overlap is high. DR-GGAD instead *anchors* normal residuals to a global Hyper Residual center and *restores* local coherence through the Affinity Residual module, so its decision boundary remains wide even as both attributes and topology shift. The model's robustness is underscored by standard deviations below 2.5% on every dataset and by additional gains on relatively easy benchmarks like Cora and CiteSeer, confirming that dual residual centering helps in favorable settings without hurting elsewhere. Complete AUPRC results, which show the same trend, are provided in the Appendix F.2.

## 5.3 ABLATION STUDY

To quantify how each component of DR-GGAD mitigates $\mathcal{AnD}$, we disable the Hyper Residual (HR) and Affinity Residual (AR) modules individually and jointly. Table 3 summarizes the AUROC of four variants: **Backbone**: only the shared GCN encoder and residual extraction. **Backbone+HR**: adds the HR center but removes AR. **Backbone+AR**: adds AR but removes the HR center. **Full**: the complete DR-GGAD with both HR and AR.

**Key findings. (1)HR addresses feature divergence.** On *Amazon* and *CiteSeer*, whose attribute statistics differ most from every source graph, adding HR lifts AUROC by +19.47 and +36.79 points, respectively. By contracting within-class residuals toward a domain-invariant center, HR

reduces $d^{(+)} + d^{(-)}$ and thus $\mathcal{A}n\mathcal{D}^*$ via Eq. equation 2, aligning with the observed AUROC gains. **(1)AR addresses topology divergence.** *Cora* and *Facebook* undergo the strongest topology shift; replacing HR with AR yields the larger boost ($+45.23$ and $+32.59$ points). Enforcing local residual alignment enlarges $d^{(+-)}$, which (by Lemma 2) relaxes the Lipschitz ceiling on score separation and, empirically, correlates with higher AUROC. **(2)HR and AR are complementary.** The full model outperforms each single-module variant by non-trivial margins (e.g., $+26.36$ on *Amazon* vs. Backbone+AR, $+16.78$ on *ACM* vs. Backbone+HR). Acting on both sides of equation 2—shrinking $d^{(+)}+d^{(-)}$ and enlarging $d^{(+-)}$—drives $\mathcal{A}n\mathcal{D}^*$ down further and yields consistently higher AUROC. **(3)Stable improvements.** Standard deviations remain under $3\%$ across five seeds, indicating robust gains rather than lucky initializations. The trend is consistent across datasets and fusion weights $\lambda$ (Figure 3).

Table 3: AUROC (%) of DR-GGAD variants. Best per row in **bold**.

| Dataset | Backbone | +HR | +AR | **Full** |
|---|---|---|---|---|
| Facebook | 49.42 | 65.30 | 82.01 | **82.16** |
| Amazon | 68.57 | 88.04 | 61.79 | **88.15** |
| ACM | 53.02 | 74.39 | 90.85 | **91.17** |
| CiteSeer | 53.87 | 90.66 | 94.03 | **95.00** |
| Cora | 47.43 | 84.77 | 92.66 | **93.20** |
| Weibo | 47.02 | 88.61 | 88.35 | **93.31** |
| BlogCatalog | 60.29 | 74.31 | 74.43 | **75.06** |
| Reddit | 50.56 | 56.04 | 58.04 | **60.60** |

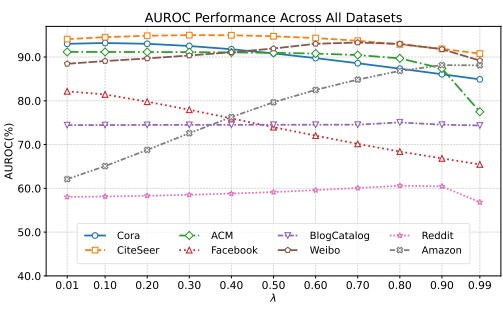

Figure 3: AUROC versus fusion weight $\lambda$ on all target graphs.

## 5.4 FUSION WEIGHT SENSITIVITY

We vary the fusion weight $\lambda$ in Eq. (13) from 0 to 1 and plot AUROC on all targets in Figure 3. Two clear regimes emerge: **(1)HRS-dominant ($\lambda > 0.6$).** *Amazon*, *BlogCatalog*, *Reddit*, and *Weibo* reach their maxima here. These graphs suffer mainly from *attribute* shift, so relying more on the Hyper Residual score—whose HR center contracts feature-space overlap—best mitigates $\mathcal{A}n\mathcal{D}$. **(2)ARS-dominant ($\lambda \approx 0.2$).** *Cora*, *CiteSeer*, *ACM*, and *Facebook* peak when the Affinity Residual score has higher weight. Their performance is limited by *topology* drift; ARS restores local structural coherence, aligning with the design goal of the AR module.

**Takeaway.** Figure 3 confirms that dual residual centering offers a tunable trade-off: HRS counters feature-level divergence, ARS counters structural divergence, and $\lambda$ lets practitioners bias the detector toward the dominant source of $\mathcal{A}n\mathcal{D}$ in a given graph—exactly as intended by our framework. An extended analysis on $\lambda$ is available in Appendix F.1.

Table 4: Relationship between residual overlap ($\mathcal{A}n\mathcal{D}^*$) and AUROC. Columns are sorted by our improvement $\Delta$. **Best** and Second AUROC are color-coded.

| Dataset | Facebook | Amazon | Cora | CiteSeer | Reddit | BlogCatalog | ACM | Weibo |
|---|---|---|---|---|---|---|---|---|
| $\mathcal{A}n\mathcal{D}^*$ Score | 0.5272 | 0.5163 | 0.5190 | 0.5294 | 0.5964 | 0.4408 | 0.4159 | 0.2745 |
| $\mathcal{A}n\mathcal{D}$ Score | 0.7850 | 0.7512 | 0.7596 | 0.7919 | 1.0000 | 0.5166 | 0.4393 | 0.0000 |
| DOMINANT | 51.01 | 48.94 | 66.53 | 69.47 | 50.05 | 74.25 | 70.08 | 92.88 |
| GCTAM | 69.57 | 55.74 | 58.78 | 70.31 | 59.32 | 67.60 | 81.21 | 70.61 |
| AnomalyGFM | 58.64 | 60.65 | 54.17 | 54.71 | 59.99 | 57.77 | 60.79 | 69.48 |
| ARC | 67.56 | 80.76 | 87.45 | 90.95 | 60.04 | 74.76 | 79.88 | 88.85 |
| Our | **82.16** | **88.15** | **93.20** | **95.00** | **60.60** | **75.06** | **91.17** | **93.31** |
| $\Delta$ | 12.59↑ | 7.48↑ | 5.75↑ | 4.05↑ | 0.56↑ | 0.30↑ | 9.96↑ | 0.43↑ |

## 5.5 IMPACT OF $\mathcal{A}n\mathcal{D}$

Table 4 juxtaposes each target's $\mathcal{A}n\mathcal{D}^*$ score with AUROC obtained by DR-GGAD and four strong baselines. Three concise observations emerge: **(1)Positive correlation with the performance gap.**

The larger the residual overlap, the larger our improvement. *Facebook* and *Amazon* possess the highest $\mathcal{A}n\mathcal{D}^*$ (0.5272 and 0.5163) and see gains of +12.59 % and +7.48 %, respectively. **(2)Collapse of prior art under extreme overlap.** When $\mathcal{A}n\mathcal{D}^* > 0.5$ (*Facebook*), ARC, DOMINANT, and GCTAM lose up to 20 AUROC points, whereas DR-GGAD still surpasses 80 %. Dual residual centering prevents the decision boundary from vanishing, confirming our design motivation. **(3)Diminishing headroom when overlap is small.** On *Weibo* ($\mathcal{A}n\mathcal{D}^* = 0.2745$) and *BlogCatalog* (0.4408), all models already separate anomalies well, so the attainable gain is modest (+0.43 % and +0.30 %). This ceiling effect is consistent with Proposition 1, which states that AUC saturates as $\mathcal{A}n\mathcal{D} \to 0$.

**Summary.** Higher $\mathcal{A}n\mathcal{D}^*$ leads to wider gaps between DR-GGAD and prior methods, while lower overlap narrows the gap—direct empirical evidence that dual residual centering neutralizes anomaly non-discriminativity in both feature and structure spaces, fulfilling the paper's core objective without any target-domain tuning.

## 6 LIMITATION

Despite its strong empirical results, **DR-GGAD is not a panacea**. We summarize the most salient limitations, each pointing to a clear avenue for future work. **(1)Weak anomaly signals.** On graphs where anomalies barely deviate (e.g., *Reddit*), residuals overlap and margins shrink. Injecting subgraph-level reconstruction or contrastive amplification could surface such subtle outliers. **(2)Sparse or noisy attributes.** Missing or adversarial features weaken the Hyper-Residual branch. A structure-only fallback or lightweight denoising encoder would improve robustness. **(3)Scalability.** Global clustering and wide residual vectors add memory and pre-processing costs; hierarchical or quantized centers can reduce this footprint on billion-node graphs. **(4)Fusion weight** $\lambda$. Choosing $\lambda$ still needs minimal validation. An unsupervised, drift-aware estimator would remove this final knob.

**Outlook.** These limits reflect open challenges for the entire GGAD field. By isolating and mitigating $\mathcal{A}n\mathcal{D}$ under strict zero-shot constraints, DR-GGAD offers a solid step forward while outlining clear paths for future research.

## 7 CONCLUSION

This paper advances representation learning for cross-domain graph anomaly detection by introducing DR-GGAD, a zero-shot framework that mitigates *Anomaly non-Discriminativity* by comparing each node to two domain-invariant residual centers. The new $\mathcal{A}n\mathcal{D}$ metric quantifies representation overlap, and the complementary *Hyper Residual* and *Affinity-Residual* modules shrink that overlap from the feature and structure perspectives. Without any target-side tuning, DR-GGAD sets a new state of the art on eight benchmarks, lifting mean AUROC by $5.14\%$ and up to $12.59\%$ on high-$\mathcal{A}n\mathcal{D}$ graphs.

## ACKNOWLEDGEMENTS

This paper is the result of the research project funded by the National Natural Foundation of China (Grant No. 62106216, 62362068, and 62162064), the Open Foundation of Yunnan Key Laboratory of Software Engineering under Grant No.2023SE104 and No.2023SE208, the Scientific Research and Innovation Project for Postgraduate Students of Yunnan University (Grant No. KC-252511575), and the Project of Yunnan Provincial Department of Education Science Research Fund (Grant No. 2026Y0106).

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

APPENDIX

# A    EXPERIMENTAL ANALYSIS OF $\mathcal{A}n\mathcal{D}$

Table 5: $\mathcal{A}n\mathcal{D}$ scores of source domain and target domain.

| Source Domain | | | | | |
|---|---|---|---|---|---|
| Dataset | PubMed | Flickr | Questions | YelpChi | Average |
| $\mathcal{A}n\mathcal{D}^*$ Score | 0.4061 | 0.4389 | 0.4079 | 0.4197 | 0.4181 |
| $\mathcal{A}n\mathcal{D}$ Score | 0.0000 | 1.0000 | 0.0592 | 0.4473 | 0.3766 |

| Target Domain | | | | | | | | | |
|---|---|---|---|---|---|---|---|---|---|
| Dataset | Facebook | Amazon | Cora | CiteSeer | Reddit | BlogCatalog | ACM | Weibo | Average |
| $\mathcal{A}n\mathcal{D}^*$ Score | 0.5272 | 0.5163 | 0.5190 | 0.5294 | 0.5964 | 0.4408 | 0.4159 | 0.2745 | 0.4774 |
| $\mathcal{A}n\mathcal{D}$ Score | 0.7850 | 0.7512 | 0.7596 | 0.7919 | 1.0000 | 0.5166 | 0.4393 | 0.0000 | 0.6305 |

To further understand how DR-GGAD alleviates Anomaly non-Discriminativity ($\mathcal{A}n\mathcal{D}^*$) under domain shifts, We perform a dual analysis: we first examine the unnormalized $\mathcal{A}n\mathcal{D}^*$ and normalized $\mathcal{A}n\mathcal{D}$ scores to quantify embedding overlap across domains; Building upon the results shown in Table 5, we proceed to analyze the role of the dual residual centering mechanism under different $\mathcal{A}n\mathcal{D}$ problem. $\mathcal{A}n\mathcal{D}^*$ score quantifies the absolute overlap between normal and anomalous embeddings within a single graph, whereas $\mathcal{A}n\mathcal{D}$ rescales these scores to the range $[0, 1]$ to enable cross-domain comparability. These metrics provide a systematic characterization of the challenges in cross-domain separability.

## A.1    $\mathcal{A}n\mathcal{D}$ SCORE ANALYSIS

**Cross-domain $\mathcal{A}n\mathcal{D}$ problem Comparison.** In the source domains (*PubMed, Flickr, Questions, YelpChi*), $\mathcal{A}n\mathcal{D}^*$ remains at a moderate level (0.4061–0.4389), with an average $\mathcal{A}n\mathcal{D}^*$ score of 0.4181. This indicates that although source domain training alleviates $\mathcal{A}n\mathcal{D}$ problem to some extent, feature overlap remains prevalent. In contrast, the average $\mathcal{A}n\mathcal{D}^*$ of target domains rises sharply to 0.4774. In particular, Reddit ($\mathcal{A}n\mathcal{D}^* = 0.5964$) and CiteSeer ($\mathcal{A}n\mathcal{D}^* = 0.5294$) exhibit severe entanglement between normal and anomalous embeddings, underscoring the amplification of $\mathcal{A}n\mathcal{D}$ problem under cross-domain transfer.

**High- $\mathcal{A}n\mathcal{D}^*$ domains ($\mathcal{A}n\mathcal{D}^* > 0.5$).** *Facebook* (0.5272), *Amazon* (0.5163), *Cora* (0.5190), *CiteSeer* (0.5294), and *Reddit* (0.5964) exhibit pronounced embedding overlap, where anomalies are nearly indistinguishable from normal nodes, making detection highly challenging.

- *Facebook/Reddit:* densely connected social networks with complex community structures; anomalous nodes often mimic normal interaction patterns, aggravating feature overlap.

- *Amazon:* e-commerce review network with high-dimensional sparse attributes, where anomalous behaviors are easily confounded with normal user ratings.

- *Cora/CiteSeer:* citation networks with highly similar semantic distributions; cross-domain transfer further introduces label semantics drift, reducing separability.

**Low-$\mathcal{A}n\mathcal{D}^*$ domains ($\mathcal{A}n\mathcal{D}^* < 0.5$).** *Weibo* (0.2745) and *ACM* (0.4159) yield relatively low scores, indicating that anomalies in these datasets remain more distinguishable and cross-domain detection is comparatively easier.

- *Weibo:* anomalies exhibit social behavior patterns that differ significantly from normal users, making anomalous nodes more salient in attributes and connectivity.

- *ACM:* as a co-authorship and publication network, anomalies often correspond to atypical collaboration or citation links, which stand out against the otherwise community-driven structure.

## A.2 The Solution of $\mathcal{A}n\mathcal{D}$ Challenge.

To address the Anomaly non-Discriminativity ($\mathcal{A}n\mathcal{D}$) problem across domains, DR-GGAD introduces a dual residual centering that integrates the Hyper Residual(HR) and Affinity Residual(AR). Empirical evidence (Table 4) shows that the level of $\mathcal{A}n\mathcal{D}^*$ strongly relates to the contributions of HR and AR. In high-$\mathcal{A}n\mathcal{D}$ domains such as *Facebook* ($\mathcal{A}n\mathcal{D}^* = 0.5272$) and *Amazon* (0.5163), where residual overlap is severe, both components are indispensable: HR enhances semantic separability, while AR captures structural irregularities. In contrast, in low-$\mathcal{A}n\mathcal{D}$ domains such as *Weibo* (0.2745) and *BlogCatalog* (0.4408), where anomalies remain comparatively more distinguishable, the score fusion strategy adaptively adjusts the weights of HR and AR, avoiding redundant complexity while maintaining stable performance. These results demonstrate that DR-GGAD flexibly adapts to varying levels of $\mathcal{A}n\mathcal{D}$ problem, achieving robust zero-shot cross-domain generalization without any target-domain tuning.

### A.2.1

## B Algorithm and Complexity

### B.1 Algorithmic Description

The overall workflow of DR-GGAD consists of a training and an inference phase (refer to Algorithm 1 and Algorithm 2). During training, the model aligns node features, extracts multi-layer GCN embeddings, and computes residuals across layers. It then jointly optimizes hyper residual clustering and affinity-based residual consistency to learn discriminative representations. In the inference phase, DR-GGAD operates without fine-tuning, leveraging pre-trained residual encoders to compute anomaly scores based on distances to hyper residual centers and local residual coherence, enabling efficient and transferable anomaly detection across graphs.

---

**Algorithm 1** Training of DR-GGAD

---

1: **Input:** Source domain graphs $\mathcal{T}_{\text{train}} = \{\mathcal{D}_{\text{train}}^{(1)}, \ldots, \mathcal{D}_{\text{train}}^{(n_s)}\}$; training epochs $E$; cluster number $\tau$.
2: **Output:** Trained model parameters; hyper residual centers $\{\bar{r}_1, ..., \bar{r}_\tau\}$.
3: **for** each graph $\mathcal{D}_{\text{train}}^{(k)} \in \mathcal{T}_{\text{train}}$ **do**
4:     Project node features into a unified representation space to obtain $\bar{\mathbf{X}}^{(i)}$.
5: **end for**
6: **for** $e = 1, \ldots, E$ **do**
7:     **for** each graph $\mathcal{D}_{\text{train}}^{(k)} \in \mathcal{T}_{\text{train}}$ **do**
8:         Compute multi-layer GCN embeddings via Eq. (5) (6)
9:         Calculate residual features $\mathbf{R}^{(i)}$ via Eq. (7)
10:         Compute and update hyper residual centers $\{\bar{r}_k\}$ using k-means Eq. (8)
11:         Compute hyper residual loss $\mathcal{L}_{HR}$ via Eq. (9)
12:         Compute affinity-residual loss $\mathcal{L}_{AR}$ via Eq. (10).
13:         Update model by minimizing joint loss $\mathcal{L} = \mathcal{L}_{HR} + \mathcal{L}_{AR}$.
14:     **end for**
15: **end for**

---

#### B.1.1 Complexity Analysis

**Training Phase.** The training time cost of DR-GGAD arises from five principal modules, each contributing to the total complexity as follows:

- **Node Attribute Alignment:** Given node features $\mathbf{X} \in \mathbb{R}^{n \times d}$, the dimensional projection to a unified space of dimension $d_u$ (e.g., via PCA or Gaussian Projection) incurs a cost of $\mathcal{O}(ndd_u)$.

- **Graph propagation and representation encoding:** For $\ell$ layers of light GCN propagation, each layer involves sparse matrix multiplications $\mathcal{O}(md_u)$, where $m$ is the number of edges. The post-layer projection into hidden space $d_h$ has cost $\mathcal{O}(nd_ud_h)$ per layer. Total complexity: $\mathcal{O}(\ell(md_u + nd_ud_h))$.

---

**Algorithm 2** Inference of DR-GGAD

---

1: **Input:** Testing graphs $\mathcal{T}_{\text{test}} = \{\mathcal{D}_{\text{test}}^{(1)}, \ldots, \mathcal{D}_{\text{test}}^{(n_t)}\}$; frozen model parameters; hyper residual centers $\{\bar{r}_1, \ldots, \bar{r}_\tau\}$; fusion weight $\lambda$.
2: **Output:** Anomaly scores $\mathcal{S}(i)$ for all test nodes.
3: **for** each graph $\mathcal{D}_{\text{test}}^{(k)} \in \mathcal{T}_{\text{test}}$ **do**
4:     Project node features into a unified representation space to obtain $\bar{\mathbf{X}}^{(i)}$.
5:     Compute multi-layer GCN embeddings via Eq. (5) (6)
6:     Calculate residual features $\mathbf{R}^{(i)}$ via Eq. (7)
7:     Calculate the hyper residual score $\mathcal{HRS}(i)$ of node $i$ via Eq. (11).
8:     Compute the affinity-residual score $\mathcal{ARS}(i)$ of node $i$ via Eq. (12)
9:     Fuse scores to obtain final anomaly score $\mathcal{S}(i)$ of node $i$ via Eq. (13)
10:     Predict node anomalies via Eq. (14).
11: **end for**

---

- **Residual computation:** Residuals are calculated via pairwise subtraction across $\ell$ layers and concatenated, leading to cost $\mathcal{O}(n(\ell - 1)d_h)$.

- **Hyper residual clustering and updating:** K-means clustering is applied to the normal node residuals to form $\tau$ centers in $\mathbb{R}^{(\ell-1)d_h}$, requiring $\mathcal{O}(n\tau(\ell - 1)d_h)$ per iteration (assuming a constant number of iterations).

- **Affinity-residual computation:** Cosine similarity is computed between each node and its neighbors in residual space, yielding $\mathcal{O}(n\bar{d}(\ell - 1)d_h)$, where $\bar{d}$ is the average node degree.

Hence, the overall training complexity becomes:

$$\mathcal{O}\Big(ndd_u + \ell(md_u + nd_ud_h) + n(\ell - 1)d_h + n\tau(\ell - 1)d_h + n\bar{d}(\ell - 1)d_h\Big) \tag{15}$$

In our experiments, the total training time across all source-domain graphs was approximately 46 seconds under a single seed (60 epochs, $\ell = 3$, $\tau = 4$), measured on a single NVIDIA RTX A6000 GPU.

**Testing Phase.** The inference complexity includes aligned projection, feature encoding, residual generation, and score computation:

- **Node Attribute Alignment:** Same as training, projection into $d_u$-dimensional space: $\mathcal{O}(ndd_u)$.

- **Embedding and residual computation:** Multi-layer GCN encoding and residual computation: $\mathcal{O}(\ell(md_u + nd_ud_h) + n(\ell - 1)d_h)$.

- **Hyper residual scoring:** Distance computation to $\tau$ pre-trained hyper residual centers: $\mathcal{O}(n\tau(\ell - 1)d_h)$.

- **Affinity-residual scoring:** Same as training: $\mathcal{O}(n\bar{d}(\ell - 1)d_h)$.

Therefore, the total inference complexity is:

$$\mathcal{O}\Big(ndd_u + \ell(md_u + nd_ud_h) + n(\ell - 1)d_h + n\tau(\ell - 1)d_h + n\bar{d}(\ell - 1)d_h\Big) \tag{16}$$

In our experiments, DR-GGAD completes inference on each target-domain graph in approximately 0.35 seconds on RTX A6000 without any fine-tuning.

To further evaluate the computational feasibility of DR-GGAD, we compare its computational cost and detection performance against representative GGAD baselines, as shown in Table 6. The key findings are as follows:

1. **Training Time and Memory:** DR-GGAD requires more training time and memory than ARC (46.06s vs. 10.26s; 35.58GB vs. 8.18GB), but the additional cost is justified by improved performance, while achieving the highest average AUROC (84.83), particularly excelling in high $\mathcal{AnD}$ domains like Facebook and Amazon.

2. **GPU Memory Usage:** UNPrompt and AnomalyGFM use more GPU memory (46.45GB and 47.42GB, respectively) compared to DR-GGAD (35.58GB), with DR-GGAD outperforming both methods.

Table 6: Comparison of computational cost and detection performance across representative GGAD methods. Average AUROC(%) is computed as the mean across 8 target datasets.

| Metric | UNPrompt | AnomalyGFM | ARC | Our |
|---|---|---|---|---|
| Training Time(s) | 9.62 | 299.19 | 10.26 | 46.06 |
| Inference Time(s) | 76.32 | 0.58 | 0.26 | 0.35 |
| GPU Memory(GB) | 46.45 | 47.42 | 8.18 | 35.58 |
| Average AUROC(%) | 56.07 | 53.28 | 78.77 | **84.83** |

3. **Inference Speed:** DR-GGAD maintains competitive inference speed (0.35s vs. 0.26s for ARC), ensuring efficient testing despite higher training costs.

4. **Future Optimizations:** Future optimizations, like parallel computing, can further enhance both training and inference speed.

In conclusion, DR-GGAD offers a practical and efficient solution with significant performance gains, making it a worthwhile trade-off in cross-domain anomaly detection.

## C   DESCRIPTION OF DATASETS

Following the experimental protocol of ARC Liu et al. (2024), Tang et al. (2023), we benchmark DR-GGAD on a suite of twelve graph datasets, organised into four domain–specific categories:

- **Citation Networks:** Injected Anomalies
- **Social Networks:** Injected Anomalies
- **Social Networks:** Real-World Anomalies
- **Co-Review Networks:** Real-World Anomalies

Within each category, the largest graph is designated as the *source* dataset for training, while the remaining graphs serve as *target* datasets for testing. This *train-on-largest, test-on-rest* paradigm imposes a stringent evaluation of DR-GGAD's cross-graph generalisation capacity. Table 7 summarises the principal statistics of all datasets. Spanning diverse application domains, the collection comprises both synthetic and naturally occurring anomalies, thereby exposing DR-GGAD to a broad spectrum of abnormal patterns during evaluation—an essential prerequisite for assessing robustness on unseen graphs. Detailed dataset descriptions are provided as follows:

- **Cora**, **CiteSeer**, **PubMed** Sen et al. (2008), and **ACM** Tang et al. (2008) are four representative citation network datasets, where nodes denote academic papers and edges represent citation links, covering diverse domains such as computer science and biomedicine. Node features are constructed using bag-of-words representations derived from textual content, and the structural and semantic variations across datasets facilitate a comprehensive evaluation of a model's cross-domain generalization capability.

- **BlogCatalog** and **Flickr** Ding et al. (2019); Tang & Liu (2009) are representative social network datasets, where nodes represent users and edges represent their social connections. Node features are derived from user-generated content such as blog texts or image tags, offering rich semantics and structural diversity for evaluating model performance on complex graphs.

- **Amazon** and **YelpChi** Rayana & Akoglu (2015); McAuley & Leskovec (2013) are graph-structured datasets built upon user review behaviors, primarily used for detecting deceptive activities and opinion spam. Amazon focuses on uncovering users involved in review manipulation through shared product interactions, while YelpChi targets suspicious reviews that distort business reputations. In this work, we utilize two specific graph constructions: Amazon-UPU, where users are linked via co-reviewed products, and YelpChi-RUR, where edges connect reviews authored by the same user, enabling the detection of subtle and coordinated anomalies.

- **Facebook** Xu et al. (2022) and **Reddit** Kumar et al. (2019) originate from real-world social networks and online community platforms, respectively. In the Facebook dataset, nodes represent users and edges denote friendship connections, reflecting localized community structures in human social graphs. Reddit models posts as nodes connected via user interactions, with node attributes derived from textual content; posts associated with banned users are labeled as anomalies for detecting malicious or disruptive behavior.

- **Weibo** Kumar et al. (2019) is a dataset derived from the Weibo platform, which is modeled as a heterogeneous graph consisting of users and their associated hashtags. It focuses on detecting abnormal behavior characterized by high-frequency posting within short time windows. Node features incorporate geolocation data and bag-of-words representations, enabling the identification of suspicious users based on both behavioral and textual patterns.

- **Questions** Platonov et al. (2023) dataset is collected from the Yandex Q platform, represents users as nodes, and encodes question–answer interactions within a one-year window as edges. Node features are computed by averaging FastText embeddings of user descriptions, with an additional binary indicator for missing descriptions, enabling effective modeling of semantic relations and interaction behaviors among users.

**Anomaly Injection.** For datasets with injected anomalies, we follow the strategies introduced in Ding et al. (2019); Liu et al. (2021) to generate both structural and attribute anomalies. Structural anomalies are created by injecting small cliques, each consisting of $p$ fully connected nodes labeled as anomalies. A total of $p \times q$ such nodes are generated per dataset, with $p = 15$ fixed and $q$ set to 10, 15, 20, 5, 5, and 20 for BlogCatalog, Flickr, ACM, Cora, Citeseer, and PubMed, respectively. Attribute anomalies are injected following the method in Song et al. (2007). For each target node $v_i$, we randomly sample $k = 50$ nodes from the graph and identify the one with the largest feature deviation from $v_i$, denoted $v_j$. The feature of $v_j$ is then assigned to $v_i$, i.e., $X_i \leftarrow X_j$, forming a camouflaged anomaly. The number of attribute anomalies is kept equal to that of structural anomalies for consistency.

Table 7: The statistics of datasets.

| Dataset | Train | Test | #Nodes | #Edges | #Features | Avg. Degree | #Anomaly | %Anomaly |
|---------|-------|------|--------|--------|-----------|-------------|----------|----------|
| Citation network with injected anomalies | | | | | | | | |
| Cora | - | ✓ | 2,708 | 5,429 | 1,433 | 3.90 | 150 | 5.53 |
| CiteSeer | - | ✓ | 3,327 | 4,732 | 3,703 | 2.77 | 150 | 4.50 |
| ACM | - | ✓ | 16,484 | 71,980 | 8,337 | 8.73 | 597 | 3.62 |
| PubMed | ✓ | - | 19,717 | 44,338 | 500 | 4.50 | 600 | 3.04 |
| Social network with injected anomalies | | | | | | | | |
| BlogCatalog | - | ✓ | 5,196 | 171,743 | 8,189 | 66.11 | 300 | 5.77 |
| Flickr | ✓ | - | 7,575 | 239,738 | 12,047 | 63.30 | 450 | 5.94 |
| Social network with real anomalies | | | | | | | | |
| Facebook | - | ✓ | 1,081 | 55,104 | 576 | 50.97 | 25 | 2.31 |
| Weibo | - | ✓ | 8,405 | 407,963 | 400 | 48.53 | 868 | 10.30 |
| Reddit | - | ✓ | 10,984 | 168,016 | 64 | 15.30 | 366 | 3.33 |
| Questions | ✓ | - | 48,921 | 153,540 | 301 | 3.13 | 1,460 | 2.98 |
| Co-review network with real anomalies | | | | | | | | |
| Amazon | - | ✓ | 10,244 | 175,608 | 25 | 17.18 | 693 | 6.76 |
| YelpChi | ✓ | - | 23,831 | 49,315 | 32 | 2.07 | 1,217 | 5.10 |

# D DESCRIPTION OF BASELINES

To ensure a fair and comprehensive evaluation of DR-GGAD, we compare it against sixteen representative baseline methods, covering three learning paradigms: supervised, unsupervised, and GGAD approaches. These categories represent common methodological frameworks in graph anomaly detection and include several state-of-the-art (SOTA) models, providing a broad basis for performance comparison under different assumptions and settings.

### D.1 SUPERVISED METHODS:

- **GCN** Kipf & Welling (2016) is a neural network that extends convolution operations to graph-structured data and enables efficient feature aggregation from nodes and their neighbors.

- **GAT** Veličković et al. (2017) is a graph neural network that introduces an attention mechanism to assign different weights to neighboring nodes, enabling more flexible feature aggregation.

- **BGNN** Ivanov & Prokhorenkova (2021) is a novel end-to-end architecture, integrates Gradient Boosted Decision Trees (GBDT) with Graph Neural Networks to handle graph-structured data with node features, significantly improving performance on node prediction tasks.

- **BWGNN** Tang et al. (2022) leverages spectrally and spatially localized band-pass filters; the model mitigates the right-shift phenomenon caused by anomalies, thereby enhancing its ability to identify and detect anomalous nodes.

- **GHRN** Gao et al. (2023) mitigates the heterogeneity problem in graph anomaly detection by precisely pruning inter-class edges through the capture of high-frequency signals in the graph.

### D.2 UNSUPERVISED METHODS:

- **DOMINANT** Ding et al. (2019) combines graph convolutional networks with deep autoencoders to jointly reconstruct graph structure and node attribute information, effectively enabling the detection of anomalous nodes in attributed networks.

- **CoLA** Liu et al. (2021) performs anomaly scoring by sampling contrastive instance pairs between nodes and their neighborhood substructures, combined with GNN encoding and anomaly-aware consistency evaluation.

- **HCM-A** Huang et al. (2022) formulates hop count prediction as a self-supervised task, integrates local and global information, and performs anomaly scoring based on prediction error and uncertainty.

- **TAM** Qiao & Pang (2023) enhances anomaly detection and homophily modeling by leveraging local affinity scoring and truncated graph optimization to mitigate the interference of heterophilous edges.

- **CAGAD** Xiao et al. (2024) a method that combines counterfactual data augmentation with a graph pointer network to identify potential anomalies and construct abnormal neighborhoods, thereby generating more distinguishable node representations.

- **Semi-GGAD** Qiao et al. (2024) is a semi-supervised anomaly detection method that generates pseudo-anomalous nodes with prior abnormal characteristics and leverages a small set of normal nodes to train a one-class classifier.

- **SmoothGNN** Dong et al. (2025) leverages the unsmoothable nature of anomaly nodes to build a multi-level modeling framework and anomaly scoring strategy, effectively distinguishing anomaly nodes from normal ones.

- **GCTAM** Zhang et al. (2025) adopts an anomaly truncation method that integrates contextual mechanisms with global affinity, effectively overcoming the misjudgment issues caused by fixed-threshold truncation in traditional methods.

### D.3 GGAD METHODS:

- **UNPrompt** Niu et al. (2025) proposes a zero-shot graph anomaly detection method based on neighborhood prompts, achieving cross-graph generalization with a single model through attribute predictability and cross-graph alignment.

- **AnomalyGFM** Qiao et al. (2025) introduces a foundation model for graph anomaly detection, which aligns residuals with prototypes through pre-training to enable cross-graph generalization under zero-shot and few-shot settings.

- **ARC** Liu et al. (2024) is the first generalist graph anomaly detection method that supports few-shot inference, achieving cross-graph generalization without retraining through feature alignment, residual encoding, and context-aware scoring modules.

# E DETAILS OF IMPLEMENTATION

## E.1 HYPER-PARAMETERS

We present the key hyperparameter settings of DR-GGAD, selected to ensure stable training and robust cross-domain generalization. The configurations below summarize the specific values used in our experiments.

- Hidden layer dimension: $\{64, 128, 265, 512, 1024\}$
- Number layer of Graph Encoder: $\{1, 2, 3, 4\}$
- Dropout rate: $\{0, 0.1, 0.2, 0.3, 0.4, 0.5, 0.6, 0.7, 0.8, 0.9\}$
- Learning rate: floats between $10^{-5}$ and $10^{-2}$
- Weight decay floats between $10^{-6}$ and $10^{-3}$
- Number of hyper residual center $\tau$: $\{2, 4, 6, 8, 12, 16, 20\}$

## E.2 EVALUATION METRICS

We evaluate model performance using two widely adopted metrics: Area Under the Receiver Operating Characteristic Curve (AUROC) and Area Under the Precision-Recall Curve (AUPRC). AUROC reflects the model's overall discriminative capability, while AUPRC is particularly informative under class imbalance. Higher values indicate better performance. All results are reported as the mean and standard deviation over five independent runs.

## E.3 EXPERIMENTAL ENVIRONMENT

All experiments were executed on a Linux server configured with Ubuntu 20.04. The system was powered by a 13th Gen Intel(R) Core(TM) i7-12700 CPU, 64GB RAM, and an NVIDIA GeForce RTX A6000 GPU with 48GB of memory. The software environment was managed using Anaconda3, with PyCharm as the development interface. Experiments were conducted under Python 3.8.14, CUDA 11.7, and PyTorch 2.0.1 Paszke et al. (2019).

## E.4 MULTI-GRAPH TRAINING PIPELINE

To ensure that the model learns residual patterns that generalize across source graphs, we adopt a coordinated multi-graph training pipeline. First, all source-domain graphs are projected into a unified feature space using PCA, which provides consistent input dimensionality and reduces dataset-specific noise. During training, graphs are processed sequentially within each epoch: for every source graph, the model performs an independent forward pass, computes the residual-based loss, and executes one optimizer update. As a result, an epoch contains multiple updates—each driven by a different graph domain.

This alternating optimization strategy prevents the model from overfitting to any single graph and gradually guides the parameters toward a domain-invariant region that is compatible with diverse structural and attribute distributions. Through repeated cross-domain updates, both the encoder and the residual modules learn residual deviation patterns that are stable across datasets, forming the basis for reliable zero-shot anomaly detection on unseen graphs.

# F SUPPLEMENTARY EXPERIMENTS

## F.1 FUSION WEIGHT ANALYSIS

We conduct a systematic evaluation of the fusion weight $\lambda \in [0.01, 0.99]$ to investigate the collaborative effect between the proposed Hyper Residual (HR) and Affinity-Residual (AR) modules

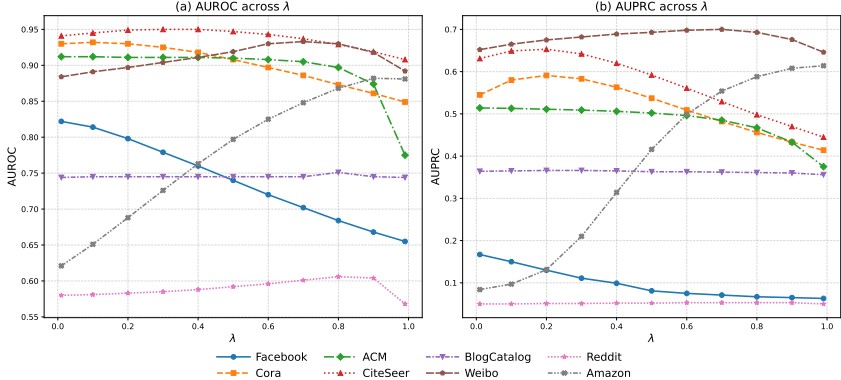

Figure 4: AUROC and AUPRC scores under different values of $\lambda$.

Table 8: **Anomaly detection performance (AUPRC, %, mean ± std).** "Rank" is the average rank over eight targets; The best, second, and third results are colored first, second, and third, respectively. Results tagged "*" are reproduced with the authors' code and default configures; all others are copied from ARC's Liu et al. (2024) leader-board.

| Method | Facebook | ACM | Amazon | Cora | CiteSeer | Reddit | Weibo | BlogCatalog | Rank |
|---|---|---|---|---|---|---|---|---|---|
| GAD Methods | | | | | | | | | |
| GCN(2017) | 1.59±0.11 | 5.27±1.12 | 6.96±2.04 | 7.41±1.55 | 6.40±1.40 | 3.39±0.39 | 67.21±15.20 | 7.44±1.07 | 11.75 |
| GAT(2018) | 3.14±0.37 | 4.70±0.75 | 15.74±17.85 | 6.49±0.84 | 5.58±0.62 | 3.73±0.54 | 33.34±9.80 | 12.81±2.08 | 9.38 |
| DOMINANT(2019) | 2.95±0.06 | 15.59±2.69 | 6.11±0.29 | 12.75±0.71 | 13.85±2.34 | 3.49±0.44 | 81.47±0.22 | 35.22±0.87 | 6.50 |
| BGNN(2021) | 3.81±2.12 | 3.48±1.33 | 7.51±0.58 | 4.90±1.17 | 3.91±1.01 | 3.52±0.50 | 30.26±29.98 | 5.73±1.47 | 12.88 |
| BWGNN(2022) | 2.54±0.63 | 7.14±0.20 | 13.12±11.82 | 7.25±0.80 | 6.35±0.73 | 3.69±0.81 | 12.13±0.71 | 8.99±1.12 | 11.00 |
| GHRN(2023) | 2.41±0.62 | 5.61±0.71 | 7.54±2.01 | 9.56±2.40 | 7.79±2.01 | 3.24±0.33 | 28.53±7.38 | 10.94±2.56 | 10.63 |
| CoLA(2021) | 1.90±0.68 | 7.31±1.45 | 11.06±4.45 | 11.41±3.51 | 8.33±3.73 | 3.71±0.67 | 7.59±3.26 | 6.04±0.56 | 10.75 |
| HCM-A(2022) | 2.08±0.60 | 4.01±0.61 | 5.87±0.07 | 5.78±0.76 | 4.18±0.75 | 3.18±0.23 | 21.91±11.78 | 6.89±0.34 | 14.63 |
| TAM(2023) | 8.40±0.97 | 23.20±2.36 | 10.75±3.10 | 11.18±0.75 | 11.55±0.44 | 3.94±0.13 | 16.46±0.09 | 10.57±1.17 | 7.25 |
| CAGAD(2024)* | 2.61±0.76 | 7.97±4.67 | 3.49±0.73 | 5.31±3.20 | 3.85±1.60 | 13.56±18.91 | 20.95±18.34 | 6.40±3.06 | 11.88 |
| Semi-GGAD(2024)* | 3.44±1.05 | 9.72±2.18 | 7.37±0.98 | 4.65±0.50 | 3.72±0.30 | 4.25±0.21 | 52.42±4.13 | 22.95±6.54 | 10.00 |
| SmoothGNN(2025)* | 2.52±0.34 | 11.60±3.32 | 20.35±17.96 | 7.23±1.96 | 5.68±1.71 | 3.40±0.24 | 28.68±20.68 | 18.64±5.17 | 9.25 |
| GCTAM(2025)* | 9.61±0.96 | 48.09±0.28 | 13.71±0.11 | 9.52±0.69 | 10.29±0.28 | 4.34±0.17 | 16.87±1.31 | 27.47±0.54 | 5.63 |
| GGAD Methods | | | | | | | | | |
| UNPrompt(2025)* | 2.61±0.45 | 10.45±1.55 | 10.27±7.04 | 6.02±0.20 | 4.47±0.32 | 5.15±0.65 | 18.67±4.33 | 24.89±3.25 | 9.25 |
| AnomalyGFM(2025)* | 5.48±3.43 | 5.41±0.69 | 11.02±5.59 | 8.65±1.06 | 7.11±0.97 | 5.03±0.39 | 37.15±16.41 | 10.44±3.38 | 7.88 |
| ARC(2024) | 8.38±2.39 | 40.62±0.10 | 44.25±7.41 | 49.33±1.64 | 45.77±1.25 | 4.48±0.28 | 64.18±0.55 | 36.06±0.18 | 3.00 |
| Our | 16.77±1.36 | 51.45±1.04 | 61.37±6.34 | 58.18±1.70 | 63.54±2.13 | 5.33±0.41 | 70.06±0.65 | 36.56±0.22 | 1.25 |

across diverse graph domains. As a key hyperparameter, $\lambda$ controls the balance between HRS and ARS, thereby directly influencing the model's capacity to address the Anomaly non-Discriminativity ($\mathcal{AnD}$) problem in both feature and structure spaces.

Empirical results from AUROC and AUPRC scores reveal a consistent unimodal trend across datasets, with peak performance often concentrated along the diagonal, as shown in Figure 4. This indicates a strong alignment between the optimal $\lambda$ and the intrinsic structure-feature dominance of each graph. Specifically, graphs with strong structural homophily (e.g., **Facebook**, **Cora**) favor smaller $\lambda$ values ($\lambda < 0.3$), where AR effectively captures local affinity residuals. In contrast, graphs characterized by attribute-driven or cross-domain heterogeneity (e.g., **Amazon**, **Weibo**) benefit from larger $\lambda$ values ($\lambda \in [0.6, 0.9]$), which amplify the generalization capacity of HR in capturing transferable feature residuals. These results highlight the necessity of adaptive $\lambda$ scheduling to align with the structural or attribute dominance of the input graph, enabling optimal synergy between HRS and ARS for robust cross-domain anomaly detection.

## F.2 PERFORMANCE COMPARISON OF AUPRC

As shown in Table 8, our method achieves the highest average AUPRC across all eight target graphs, consistently outperforming all baselines. Compared to the strongest existing GGAD method, ARC, our approach yields substantial improvements on all datasets, especially on domains with severe feature-structure entanglement such as **Facebook** (+8.39%), and **Amazon** (+17.12%), demonstrating superior discriminative capability under challenging anomaly non-separability. Traditional su-

pervised models (e.g., GCN, GAT, BGNN) exhibit limited generalization due to their reliance on domain-specific labels. Even recent variants like CAGAD and GHRN show significant performance variance under structural shifts. Among unsupervised baselines, methods such as DOMINANT and GCTAM perform well on selected datasets but degrade under noisy features or heterogeneous structures. ARC performs competitively on graphs like **Cora** and **CiteSeer**, benefiting from residual contextual modeling, but suffers on structurally diverse graphs such as **Reddit** and **Amazon**. In contrast, our method integrates the strengths of the HR and AR modules: HR promotes transferable feature residual encoding, while AR enforces local structural alignment, resulting in more robust and adaptive anomaly detection. Moreover, our approach maintains low variance across datasets, indicating strong stability even on complex graphs like **BlogCatalog** and **Weibo**, and further substantiating the robustness of our framework under diverse anomaly distributions and graph structures.

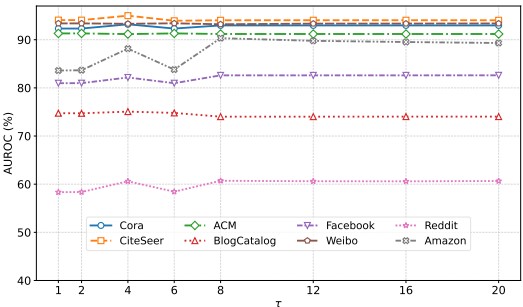

Figure 5: Sensitivity analysis of $\tau$

## F.3 SENSITIVITY ANALYSIS OF HR CENTERS

As illustrated in Figure 5, DR-GGAD exhibits a clear and consistent sensitivity pattern with respect to the number of residual centers $\tau$. When $\tau$ increases from 1 to around 4–6, most datasets show a noticeable AUROC improvement, suggesting that normal nodes indeed occupy multiple residual modes and that a small set of prototypes is sufficient to characterize their multi-scale variation. Beyond this range, however, the performance quickly stabilizes: for $\tau \geq 8$, all curves enter a plateau with fluctuations generally below 0.5%, indicating that the residual structure is already well captured and additional centers only over-segment existing clusters without introducing new information. This stability across eight diverse graphs demonstrates the robustness of the prototype mechanism and supports our choice of $\tau = 4$, which provides strong performance while avoiding unnecessary redundancy and computational overhead.

## F.4 ABLATION VISUALIZATION

To validate the contribution of different modules to anomaly detection performance, we conducted ablation experiments with histogram visualizations on the ACM and Weibo datasets, comparing the anomaly score distributions of the AR module, HR module, and the complete DR-GGAD model. The results show that AR and HR modules have complementary advantages in detecting different types of anomalies, as shown in Figures 6 to 11. (1) On the ACM dataset, the AR module effectively distinguishes between normal and anomalous nodes, with a small overlap area of 0.0490, demonstrating strong structural anomaly detection capability. In contrast, the HR module shows weaker discrimination, with an overlap area of 0.6016. When combined, the complete model reduces the overlap area further to 0.0287, significantly enhancing anomaly separability and showcasing the performance improvement from module synergy. (2) On the Weibo dataset, due to more complex features, the AR module performs significantly worse (overlap area of 0.6283), and the HR module also struggles to distinguish normal from anomalous nodes (overlap area of 0.2803). However, the complete model reduces the overlap area to 0.2124, demonstrating that dual-residual modeling can still effectively restore good anomaly separability even in complex scenarios. In summary, the AR and HR modules each excel at detecting structural and feature-based anomalies, respectively. Their collaboration significantly reduces the overlap between normal and anomalous nodes, allevi-

ating the Anomaly non-Discriminativity ($\mathcal{AnD}$) problem and improving the model's cross-domain generalization ability.

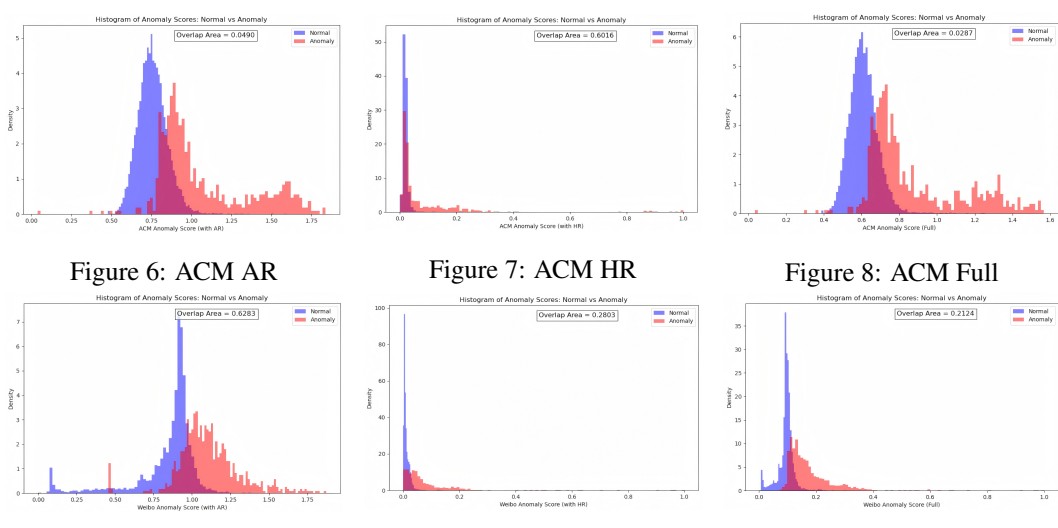

| Figure 6: ACM AR | Figure 7: ACM HR | Figure 8: ACM Full |
|---|---|---|
| Figure 9: Weibo AR | Figure 10: Weibo HR | Figure 11: Weibo Full |

These findings confirm that multi-layer residuals provide complementary structural and semantic signals, and the concatenation strategy significantly improves anomaly detection performance.

Table 9: Comparison of K-Means and Spectral Clustering with AUROC.

| Method | Facebook | ACM | Amazon | Cora | CiteSeer | Reddit | Weibo | BlogCatalog | Time(s) |
|---|---|---|---|---|---|---|---|---|---|
| K-means | 50.06 | 72.77 | 57.75 | 57.71 | 55.44 | 41.85 | 93.90 | 72.81 | - |
| Our (Spectral Clustering) | 82.53 | 91.11 | 88.14 | 93.14 | 95.06 | 60.59 | 93.25 | 74.08 | 1239.0 |
| Our (K-means) | 82.16 | 91.17 | 88.15 | 93.20 | 95.00 | 60.60 | 93.31 | 75.06 | 5.36 |

### F.5 COMPARISON OF K-MEANS AND SPECTRAL CLUSTERING FOR HR CENTER CONSTRUCTION

To compare the performance of K-means and Spectral Clustering in constructing Hyper Residual (HR) centers, we conducted an in-depth analysis of their differences in terms of performance, computational feasibility, and scalability.

**Method Difference:** K-means directly minimizes the Euclidean distance in the residual feature space, making it a distance-driven clustering method. On the other hand, Spectral Clustering requires constructing the graph Laplacian and performing eigen-decomposition to map the nodes into the spectral space before applying K-means. It emphasizes structural information but comes with higher computational costs.

**Experimental Results:** As shown in Table 9, both K-means and Spectral Clustering achieve nearly identical performance in the HR module. The differences in AUROC are minimal (e.g., Cora: 93.20 vs. 93.14, ACM: 91.17 vs. 91.11), indicating that both methods capture the same dominant normal modes in the multi-scale residual space.

**Computational Cost:** Spectral Clustering requires eigen-decomposition of the $n \times n$ Laplacian matrix, which has a computational complexity of **O(n³)**, resulting in much higher running time than K-means. In our experiments, spectral clustering took **1239 seconds**, while K-means took only **5.36 seconds**. Moreover, as the graph size increases, the stability of spectral clustering declines, making it unsuitable for large-scale or cross-domain scenarios. Despite similar performance, Spectral Clustering is computationally expensive and lacks scalability for large-scale datasets. Therefore, K-means is preferred due to its stability, efficiency, and practicality in constructing the Hyper Residual centers.

## G  Discussion

### G.1  Analysis of $\mathcal{A}n\mathcal{D}$ vs. MMD in Anomaly Detection

To investigate the relationship between class overlap and anomaly detection performance, we compare our proposed Anomaly non-Discriminativity ($\mathcal{A}n\mathcal{D}$) with Maximum Mean Discrepancy (MMD). While both metrics measure distributional differences, they capture distinct aspects of the embedding space. $\mathcal{A}n\mathcal{D}$ quantifies the class-conditional overlap between normal and anomalous nodes, reflecting the difficulty of distinguishing these two classes within a single embedding space. In contrast, MMD measures the global distribution shift between two sets of embeddings, but it does not directly capture class separability.

Empirically, we analyze Inter-domain MMD, Intra-domain MMD, and $\mathcal{A}n\mathcal{D}^*$ across eight datasets, correlating them with AUROC (Table 10). Inter-domain MMD measures the distribution shift between different domains (source and target domains), while Intra-domain MMD measures the distribution shift within the same domain, specifically between normal and anomalous nodes. $\mathcal{A}n\mathcal{D}^*$ shows the highest Pearson correlation with AUROC (0.7508), far surpassing MMD, which has only a moderate or weak correlation. For example, on the Amazon dataset, Intra-domain MMD is high, but AUROC is low, with MMD failing to capture this discrepancy. However, $\mathcal{A}n\mathcal{D}^*$ reaches 0.5163, accurately reflecting the class overlap and aligning with the observed performance decline.

These results highlight that MMD and $\mathcal{A}n\mathcal{D}^*$ measure orthogonal concepts: global distribution shift and class separability. Since anomaly detection performance is driven by separability, $\mathcal{A}n\mathcal{D}^*$ provides a more robust and consistent indicator.

Table 10: Comparison of distribution-shift metrics and the correlation with AUROC.

| Dataset | Cora | CiteSeer | ACM | BlogCatalog | Facebook | Weibo | Reddit | Amazon | Correlation |
|---|---|---|---|---|---|---|---|---|---|
| **Inter-domain MMD** | 0.8545 | 1.0002 | 0.5933 | 0.5421 | 0.8077 | 0.9478 | 0.5189 | 0.3416 | -0.7246 |
| **Intra-domain MMD** | 0.0115 | 0.0092 | 0.0454 | 0.1911 | 0.0382 | 0.1362 | 0.0002 | 0.1758 | -0.2302 |
| $\mathcal{A}n\mathcal{D}$ | 0.5190 | 0.5294 | 0.4159 | 0.4408 | 0.5272 | 0.2745 | 0.5964 | 0.5163 | 0.7508 |
| **AUROC** | 59.64 | 60.27 | 60.49 | 58.91 | 55.68 | 70.56 | 42.02 | 45.49 | - |

### G.2  Discussion of Clustering-based Method

Clustering-based methods usually enhance anomaly detection by leveraging two core strategies: strengthening intra-cluster compactness while expanding inter-cluster separation, and aligning feature or cluster distributions to discriminate anomalous instances from normal ones better Aytekin et al. (2018), Kulatilleke et al. (2025), Sohn et al. (2023). For instance, CARE Zheng et al. (2025) integrates soft membership assignments into the adjacency matrix and uses contrastive learning to capture affinities among nodes. Similarly, discDC Cai et al. (2025) explicitly optimizes intra- and inter-cluster separability to enhance deep clustering performance, ensuring a compact within-cluster structure and clear separation between clusters. ClusterQ Gao et al. (2022) focuses on feature-distribution alignment, addressing the issue of model quantisation without access to original data. By clustering the generated data and aligning its feature distribution to mimic real data, ClusterQ improves the inter-class separability and maintains high-quality anomaly detection even in data-limited scenarios.

DR-GGAD adopts Dual Residual Centering, introducing the innovative Hyper Residual (HR) center and Affinity Residual (AR) module. Unlike existing methods such as CARE and discDC, we optimize the compactness and separability of the source domain's clustering, performing optimization in both the feature and structural spaces during training. This approach overcomes the performance degradation caused by feature overlap and structural changes in traditional methods, while simultaneously improving the robustness of cross-domain transfer. Additionally, our method eliminates the need for labels in the target domain and avoids the necessity for re-clustering, further minimizing the dependency on the target domain.

### G.3 ANALYSIS OF AR MODULE'S EFFECTIVENESS

To explore the effectiveness of the AR module in revealing structural anomalies, we conducted experimental validation.

**Design Motivation**: The AR module works by measuring the consistency of node residuals with respect to their neighborhood structure. Normal nodes typically maintain stable residual directional consistency, while structural anomalies cause significant changes in the residual direction. AR primarily focuses on:

- **Neighborhood Structure Consistency:** The alignment of a node's residual with its neighbors.

- **Topological Deviation:** Deviations caused by irregular neighborhood connections or structural anomalies that deviate from normal patterns.

As such, AR is highly sensitive to anomalies caused by topological deviations, such as motif anomalies and anomalous cross-community edges.

**Experimental Verification**: We conducted structural anomaly injection experiments based on previous work Ding et al. (2019), Liu et al. (2021), where fully connected small cliques (size $p$) were injected into anomalous nodes to simulate strong structural anomalies. As $p$ increases, the strength of the structural anomaly increases, and the experimental results are shown in Table 11:

- **Structural Perturbations:** As the structural anomaly increases (from $p = 2$ to $p = 6$), AR's AUROC improves significantly, indicating that AR has a monotonic response to topological anomalies.

- **Sensitivity to Topological Changes:** The improvements are most pronounced on datasets with relatively homogeneous structures, such as Cora, CiteSeer, and ACM, showing that AR is particularly sensitive to topological perturbations.

- **Robustness Across Diverse Datasets:** Even in datasets with high noise levels (such as Reddit and Weibo), AR still shows stable improvements, demonstrating its robustness in various topological conditions.

The experimental results show that Affinity Residual (AR) is highly responsive to structural anomalies, with its performance positively correlated with the intensity of the anomalies, especially when there are changes in the topological structure. AR's effectiveness in topology-based anomaly detection has been validated, particularly when anomalies arise primarily from changes in graph structure rather than node attributes.

Table 11: Performance (AUROC) of AR with varying structural noise injection across different datasets.

| Method | Facebook | ACM | Cora | CiteSeer | Reddit | Weibo | BlogCatalog | Amazon |
|---|---|---|---|---|---|---|---|---|
| AR-Only | 82.16 | 90.85 | 93.20 | 94.03 | 58.04 | 88.35 | 74.43 | 61.79 |
| AR + Structural noise ($p = 2$) | 82.48 | 92.17 | 94.63 | 95.19 | 61.77 | 91.21 | 74.60 | 59.03 |
| AR + Structural noise ($p = 4$) | 83.28 | 93.88 | 94.58 | 95.93 | 68.10 | 92.47 | 74.76 | 57.33 |
| AR + Structural noise ($p = 6$) | 83.62 | 94.11 | 95.32 | 96.51 | 69.84 | 92.86 | 75.42 | 57.15 |

## H DECLARATION OF LLM USAGE

We used Large Language Models (LLMs) solely as an auxiliary for polishing certain parts of the manuscript, including grammar correction and fluency improvement. LLMs had no role in the study conception, method design, experimental implementation, data analysis, or generation of scientific conclusions. All technical content, results, and conclusions in the manuscript are the sole responsibility of the authors. The authors take full responsibility for all content in this manuscript.

# I   REPRODUCIBILITY STATEMENT

We believe that reproducibility is essential for validating our research. To ensure that our work can be reproduced, we have made the following efforts:

- **Code Availability**: The source code for the model and experiments is available at the GitHub repository: https://github.com/bfrnlkj/DR-GGAD.git, covering the model's core components and main experiment flows.

- **Data Processing**: A detailed description of the data processing steps, including data cleaning, feature extraction, and transformation procedures, is included in the GitHub repository. This will help replicate the datasets used in our experiments.

- **Experiment Setup**: The experimental setup, including hyperparameter configurations, training procedures, and evaluation protocols, is described in detail in both the main text and the appendix to ensure that the experiments can be faithfully reproduced.

By making these resources publicly available and providing comprehensive explanations, we aim to support reproducibility and enable other researchers to validate and extend our work.

