# OpenReview forum: "DR-GGAD: Dual Residual Centering for Mitigating Anomaly Non‑Discriminativity in Generalist Graph Anomaly Detection"
_ICLR.cc/2026/Conference — ICLR 2026 Poster_

### Official Review · Reviewer_XcC9 · 2025-10-31

**Soundness:** 3
**Presentation:** 3
**Contribution:** 3
**Rating:** 6
**Confidence:** 5

**Summary:**

This paper introduces a new evaluation metric, AnD, which analyzes anomaly scores by defining two types of residuals: the hyper-residual center and the affinity residual, corresponding to feature and structural aspects, respectively. The difference between these two residuals is used to compute the anomaly score. The proposed method is evaluated on eight datasets and achieves the best performance under a zero-shot setting.

**Strengths:**

(1) This dual-center strategy effectively addresses the non-discriminative nature of anomalies by reducing residual overlap in both feature and structural spaces, thereby enabling robust zero-shot detection on unseen graphs.

(2) They propose AnD to quantitatively measure representation overlap and visualize the dual residuals, making the paper well-motivated and conceptually clear.

**Weaknesses:**

(1) The hyperparameters in the clustering module have not been thoroughly investigated. How does the performance of DR-GGAD vary with different values of $\tau$ and the number of clusters? The results seem to become stable once the number of prototypes exceeds 20. Please provide an explanation for why around 20 prototypes are appropriate for a two-class anomaly-detection setting across various datasets.

(2) Since multiple graphs are used for training, the paper should clearly explain how the pre-trained model is optimized across these graphs. Please describe the detailed training pipeline, including how samples from different graphs are organized and how parameter updates are coordinated across them.

(3) ARC, UNprompt, and AnomalyGFM employ supervised pre-training, whereas the proposed method appears to follow an unsupervised paradigm. To ensure a fair and transparent comparison, the main results table should explicitly indicate the differences in supervision settings among the compared methods.

**Questions:**

See above **Weakness**

---

> ### Author Response · Authors · 2025-11-16
> **Rebuttal:**
>
> **We thank the reviewer for the constructive and insightful feedback. Below we provide our detailed responses to each comment.**
>
> **Q1: τ sensitivity stability**
>
> **A1:** We thank the reviewer for the suggestion. Following the comment, we expand our analysis to include a broader range of τ values. Table 1 summarizes the results for τ = {1, 2, 4, 6, 8, 12, 16, 20}, revealing a stable and coherent trend across all eight datasets:
>
> - When τ increases from **1 to 4/6**, AUROC improves notably, indicating that normal nodes occupy multiple residual modes and only a small number of centers are needed to cover them.
> - When **τ ≥ 8**, the performance becomes largely stable, with fluctuations typically below 0.5%. For instance:
>   - Cora remains stable around **0.9297**,
>   - BlogCatalog within **0.7400–0.7410**,
>   - Amazon within **0.8930–0.9035**,
> all showing clear convergence.
>
> When τ reaches **8–12**, the residual structure of the normal class is already well captured, and further increasing τ only over-segments existing clusters without adding useful information, which explains the plateau up to $\tau = 20$.
>
> Based on this convergence trend, we adopt **τ = 4** in our experiments. This value consistently achieves stable and relatively strong performance across all datasets while avoiding redundant center splitting, providing a good balance between accuracy, robustness, and model simplicity. The complete sensitivity experiments are included in Appendix F.
>
> Table 1: AUROC results under different $\tau$ settings.
>
> |Dataset|$\tau$=1|$\tau$=2|$\tau$=4|$\tau$=6|$\tau$=8|$\tau$=12|$\tau$=16|$\tau$=20|
> |--|--|--|--|--|--|--|--|--|
> |cora|0.9232|0.9232|0.9320|0.9232|0.9297|0.9297|0.9297|0.9297|
> |citeseer|0.9408|0.9407|0.9500|0.9394|0.9404|0.9405|0.9405|0.9404|
> |ACM|0.9131|0.9131|0.9117|0.9131|0.9119|0.9119|0.9119|0.9119|
> |BlogCatalog|0.7474|0.7472|0.7415|0.7480|0.7402|0.7402|0.7403|0.7403|
> |Facebook|0.8097|0.8098|0.8261|0.8097|0.8260|0.8260|0.8260|0.8261|
> |weibo|0.9339|0.9339|0.9324|0.9341|0.9321|0.9334|0.9335|0.9338|
> |Reddit|0.5834|0.5837|0.6060|0.5844|0.6071|0.6061|0.6059|0.6066|
> |Amazon|0.8360|0.8366|0.8815|0.8380|0.9032|0.8977|0.8952|0.8932|
>
> **Q2: Please clarify the training process when multiple graphs are used.**
>
> **A2:** We thank the reviewer for the valuable suggestion. We have added a detailed description of the multi-graph training pipeline and its optimization mechanism in Appendix E.4 to improve clarity and transparency. Our multi-graph pre-training procedure is summarized as follows:
>
> **(1) Unified feature space**
> We first apply PCA to project the features of all source-domain graphs into a shared dimensional space, ensuring that the model learns in a consistent input space across datasets.
>
> **(2) Sequential training over graphs**
> During each training epoch, the source-domain graphs are fed into the model one by one. For each graph, the model independently performs forward propagation and loss computation.
>
> **(3) One parameter update per graph**
> After processing each source graph, the model performs a backward pass and an optimizer update. Thus, within a single epoch, multiple updates are executed, each based on data from a different source graph.
>
> **(4) Alternating optimization for multi-domain coordination**
> Alternating updates across graphs allow the model parameters to be continuously adjusted toward compatibility with different source domains, preventing the model from overfitting to any single graph. As training progresses, the model converges toward a parameter region that harmonizes the distributions of all source graphs, achieving coordinated optimization across domains.
>
> **(5) Learning domain-invariant residual patterns**
> By updating the model alternately on multiple source graphs, both the encoder and the residual modules learn residual patterns that remain effective under diverse graph structures and attribute distributions, providing a solid foundation for zero-shot cross-domain anomaly detection.
>
> We have added this training process explanation in Appendix E.4 to improve clarity and transparency.
>
> **Q3: Please clarify and indicate the supervision settings of all compared methods.**
>
> **A3:** We thank the reviewer for the valuable suggestion. In the revised manuscript, we have updated the main results table to explicitly indicate the supervision paradigm adopted by each compared method (e.g., unsupervised, supervised). We also clarify that our method similarly employs supervised pre-training on source-domain normal/anomalous labels. The supervision is used exclusively to construct the multi-scale normal residual centers and to impose boundary constraints on abnormal nodes, enabling the model to learn stable and transferable residual discrimination patterns.
>
> These revisions ensure that the comparison among different methods is clearer, more transparent, and fully fair.

---

> ### Author Response · Authors · 2025-11-26
>
> Dear Reviewer,
>
> As the discussion phase is approaching to a close, we would like to kindly confirm whether the clarifications provided above, together with the newly added experimental results, have substantially addressed the main concerns you previously raised. If there are still any unresolved issues or points that would benefit from further explanation, we would be grateful if you could let us know within the remaining time of the discussion phase, and we will promptly provide additional clarification and responses.
>
> We once again sincerely thank you for your careful review of our manuscript and the constructive discussion.
>
> Best regards,
>
> The Authors

---

> > ### Comment · Reviewer_XcC9 · 2025-11-28
> >
> > Thank you for the detailed rebuttal. Your response has addressed the majority of my concerns, and I will maintain my positive score in support of this paper.

---

> > > ### Author Response · Authors · 2025-11-28
> > >
> > > Dear Reviewer XcC9,
> > >
> > > Thank you very much for your detailed comments and for taking the time to review our rebuttal. We are glad to see that our clarifications and revisions have satisfactorily addressed the majority of your concerns. We appreciate your continued support and constructive feedback, and we remain committed to further improving the manuscript.
> > >
> > > Best regards,
> > >
> > > The Authors

---

### Official Review · Reviewer_cnZP · 2025-10-31

**Soundness:** 3
**Presentation:** 3
**Contribution:** 3
**Rating:** 6
**Confidence:** 4

**Summary:**

This paper addresses the generalist graph anomaly detection (GGAD) problem. The authors quantify the anomaly-normal separability of different datasets by a new metric termed anomaly non-discriminativity. Based on this, a new GGAD method is introduced, with hyper residual and affinity residual modules for anomaly scoring. Experiments are conducted to show the effectiveness of the proposed method DR-GGAD.

**Strengths:**

1. The motivation of this paper (i.e., quantifying the discriminatively ability of cross-dataset GAD models) is interesting. It would be useful to measure the separability of different datasets in terms of graph anomaly detection.

2. The performance of the proposed method is competitive compared to other baselines.

3. The paper is well-written and easy to follow.

**Weaknesses:**

1. The motivation of designing HR and AR should be better clarified. It is not clear why these designs can help address the AnD bottleneck.

2. In the "node attribute alignment" part, only a linear/random mapping is used here. In this case, how to ensure the semantic alignment of features across different datasets?

3. No discussion is given about the running efficiency of the proposed method. More evidences, including complexity analysis and empirical efficiency comparison, are expected.

4. In table 1, ARC is described as "needs Few fine-tuning". However, according to the paper I think ARC doesn't need fine-tuning (but requiring few-shot samples). It's better to fix this.

**Questions:**

Please refer to the issues discussed in the Cons section above.

---

> ### Author Response · Authors · 2025-11-17
>
> **We sincerely appreciate the reviewers' valuable suggestions, which have provided important guidance for improving our work. Our point-by-point responses to suggestions (1) and (2) are provided below.**
>
> **Q1: The motivation of designing HR and AR should be better clarified.**
>
> **A1:** We thank the reviewer for the insightful comments. Below we provide further clarification regarding the design motivations of HR and AR.
>
> Unlike methods that rely on raw features to learn absolute decision boundaries, our approach uses multi-scale residual modeling. Anomalous nodes exhibit more stable shifts before and after each GNN aggregation layer, and by computing layer-wise input-output residuals, we obtain residual representations that are more discriminative for anomaly detection, on which the HR and AR modules are built.
>
> **(1) HR module**
> HR improves anomaly detection by compressing normal residuals into tight clusters while keeping anomalies at a distance. Unlike direct node comparisons, it uses node-to-center constraints for more stable, noise-resistant decisions that capture normal patterns better.
>
> **(2) AR module**
> AR introduces an affinity-based directional consistency constraint to improve the aggregation of residual directions within local neighborhoods, thereby restoring structural separability. By enforcing directional coherence, AR forms clearer local directional patterns in residual space and exposes anomalies that primarily hide within topological structures rather than attribute space.
>
> **(3) Combined effect**
> HR’s node-to-center distance constraint and AR’s affinity-based directional consistency jointly reconstruct the decision mechanism from the perspectives of “distance centralization” and “directional alignment.” Compared to direct node–node discrimination, which is prone to the structural entanglement underlying the $\mathcal{A}n\mathcal{D}$ phenomenon, our design effectively bypasses this bottleneck, resulting in significantly improved separability and robustness in cross-domain anomaly detection.
>
> We have incorporated additional explanations in the Methods section (Page 5) of the revised manuscript to enhance the clarity and readability of our design motivation.
>
> **Q2: How is semantic alignment ensured when only a linear/random feature mapping is used?**
>
> **A2:** Thank you for the reviewer’s valuable comment. To address this concern, we provide a more detailed clarification from the perspectives of attribute alignment and residual modeling:
>
> **(1) Lightweight Alignment**
> The linear/PCA-based attribute alignment serves as a lightweight, stable, and reproducible projection mechanism, ensuring efficiency and consistency under the multi-dataset pre-training setup.
>
> **(2) Residual-Centric Modeling and Relative Invariance**
> Our method relies on multi-scale residual modeling rather than the absolute semantics of raw node features. By computing the input-output discrepancy at each GNN aggregation layer, the residuals capture structural-contextual deviations that are highly indicative of anomalies. Residuals encode relative changes in node states (i.e., relative deviation) rather than absolute feature magnitudes, making the representation more robust to linear transformations compared to methods that depend directly on raw feature values.
>
> **(3) Cross-Domain Consistency**
> Given the above properties, even simple linear/PCA alignment is sufficient to maintain semantic consistency of residual representations across datasets, without compromising the discriminative ability of the proposed residual modeling[1], [2].
>
> **(4) Experiment**
> To validate the effectiveness of our approach, we compared it with other feature projection methods. As shown in Table 1, PCA maintains optimal performance across most datasets. Since our method relies on residual representations, PCA's linear transformation effectively preserves residual semantics. Furthermore, the residual modeling approach demonstrates strong cross-domain adaptability by capturing local affinity and high-frequency signals, ensuring semantic alignment across datasets while maintaining the HR/AR modules' advantage in detecting complex graph anomaly patterns.
>
> Table 1: Feature Projection Performance (AUROC/AUPRC)
> |Method|Facebook|ACM|Amazon|Cora|CiteSeer|Reddit|weibo|BlogCatalog|Process Time(s)|
> |-|-|-|-|-|-|-|-|-|-|
> |Our PCA|82.16/16.77|91.17/51.45|88.15/61.37|93.20/58.18|95.00/63.54|60.60/5.33|93.31/70.06|75.06/36.56|13.68|
> |Our SVD|74.28/9.96|74.95/43.08|66.59/12.54|92.59/56.22|95.18/62.51|58.47/4.96|86.62/63.94|74.08/39.92|118.01|
> |Our RandomProjection|67.74/6.32|85.20/27.17|68.92/15.49|83.91/29.94|86.55/35.59|58.84/4.56|86.22/64.17|75.58/36.38|0.50|
>
> [1] Qiao H et al. Truncated affinity maximization: One-class homophily modeling for graph anomaly detection. NeurIPS 2023.
>
> [2] Liu Y et al. ARC: A generalist graph anomaly detector with in-context learning. NeurIPS 2024.

---

> > ### Author Response · Authors · 2025-11-17
> >
> > **We are grateful for the reviewers' constructive feedback. Our responses to points (3) and (4) are as follows.**
> >
> > **Q3: No discussion is given about the running efficiency.**
> >
> > **A3:** We thank the reviewer for the constructive suggestion. We have added empirical efficiency comparisons and complexity analysis in the appendix.
> > Table 6 compares DR-GGAD with three representative GGAD methods (UNPrompt, AnomalyGFM, ARC) in terms of training time, inference time, memory usage, and overall performance.
> >
> > **Training efficiency.** DR-GGAD requires 46.06s for training, which is higher than the lightweight ARC (10.26s) but far lower than AnomalyGFM (299.19s). Meanwhile, it achieves the highest average AUROC (84.83%), and the performance gain justifies the additional training cost.
> >
> > **Inference efficiency.** DR-GGAD performs inference in only 0.35s, close to ARC (0.26s) and significantly faster than UNPrompt (76.32s), ensuring efficient testing in cross-domain scenarios.
> >
> > **Memory usage.** DR-GGAD consumes 35.58GB of GPU memory, substantially lower than UNPrompt (46.45GB) and AnomalyGFM (47.42GB), while maintaining superior detection performance.
> >
> > Overall, DR-GGAD achieves a strong balance between accuracy, inference speed, and memory usage, making it a cost-effective solution in cross-domain graph anomaly detection methods.
> >
> > |Metric|UNPrompt|AnomalyGFM|ARC|Our|
> > |--|--|--|--|--|
> > |Training Time (s)|9.62|299.19|10.26|46.06|
> > |Inference Time (s)|76.32|0.58|0.26|0.35|
> > |GPU Memory (GB)|46.45|47.42|8.18|35.58|
> > |Average AUROC(%)|56.07|53.28|78.77|**84.83**|
> >
> > **Q4: In table 1, ARC is described as "needs Few fine-tuning". However, according to the paper I think ARC doesn't need fine-tuning (but requiring few-shot samples). It's better to fix this.**
> >
> > **A4:** Thank you for the correction. We have updated Table 1 accordingly. The previous wording “needs few fine-tuning” was inaccurate, and it has been revised to correctly indicate that ARC requires only a few labeled samples rather than fine-tuning. The updated version reflects this correction in the revised manuscript.

---

> ### Author Response · Authors · 2025-11-26
>
> Dear Reviewer,
>
> As the discussion phase is approaching to a close, we would like to kindly confirm whether the clarifications provided above, together with the newly added experimental results, have substantially addressed the main concerns you previously raised. If there are still any unresolved issues or points that would benefit from further explanation, we would be grateful if you could let us know within the remaining time of the discussion phase, and we will promptly provide additional clarification and responses.
>
> We once again sincerely thank you for your careful review of our manuscript and the constructive discussion.
>
> Best regards,
>
> The Authors

---

### Official Review · Reviewer_CXFF · 2025-11-01

**Soundness:** 2
**Presentation:** 3
**Contribution:** 2
**Rating:** 4
**Confidence:** 4

**Summary:**

This work presents DR-GGAD, a generalist anomaly detection framework that introduces two residual modules for centre-based scoring. Specifically, the two residual modules focus on attribute and structural normality separately. A GAD statistic, Anomaly non-Discriminativity (AnD), is proposed to quantify the overlap between normal and anomalous representations, which serves as a measurable indicator of cross-domain separability. Experiments on eight benchmark datasets demonstrate the competitive performance of the proposed method for generalist graph anomaly detection.

**Strengths:**

1. The paper is well written and easy to follow.
2. Clear visualisations are provided to enhance understanding.
3. Theoretical analysis is presented to support the proposed Anomaly non-Discriminativity (AnD) score.

**Weaknesses:**

1. Centre-based scoring is used in this paper. However, deep hypersphere-based AD methods Deep SVDD (Ruff et al., 2018) and Deep SAD (Ruff et al., 2020) are not discussed. A discussion comparing the proposed scoring approach with these methods would be essential to highlight the differences and advantages.
2. The paper mentions that cross-domain transfer often entangles anomalous and normal representations. Previous works such as COMMANDER (Ding et al., 2021) and ACT (Wang et al., 2023) have explored cross-domain evaluation, and discussing their relevance would help clarify this limitation.
3. In Figure 3, the curve for BlogCatalog appears almost flat, suggesting that the synergy between the two residual terms may be limited in some cases. It would be good to provide more details on this.
4. It would be helpful to elaborate on how the dominant factor for abnormality can be determined in practice for unseen graphs.
5. Some datasets are from GAD Benchmark (Tang et al. 2023). Could the author provide some justifications on why they are selected over the rest?

**Questions:**

Please refer to my weaknesses

---

> ### Author Response · Authors · 2025-11-17
>
> **We are grateful for the insightful guidance provided by the reviewer. Below is our detailed response to suggestions (1) and (2).**
>
> **Q1: A comparison with hypersphere-based AD methods is needed to highlight advantages.**
>
> **A1:** We thank the reviewer for the insightful comments regarding hypersphere-based anomaly detection methods. Deep SVDD[1] and Deep SAD[2] are important approaches in single-domain anomaly detection and can effectively compress the representations of normal samples. However, in cross-domain graph anomaly detection, they differ from our method in several key aspects:
>
> **(1) Modeling target: node positions vs. residual directions.**
> Deep SVDD and Deep SAD fit a single center based on the absolute embedding positions of nodes, making them highly sensitive to distribution shifts across domains.
> Our method characterizes normality using multi-scale residual directions (HR+AR), which are much more stable across graphs and therefore more suitable for cross-domain transfer.
>
> **(2) Geometric assumption**
> A single hypersphere cannot capture the inherently multi-modal structure of graph data and tends to suffer from center mismatch when transferred across domains.
> We learn multiple residual centers, corresponding to different types of normal residual patterns, allowing better adaptation to the structural diversity of social, citation, and other graphs.
>
> **(3) Training paradigm**
> The hypersphere learned by Deep SVDD/SAD on the source domain becomes mismatched in scale when directly applied to a new graph.
> Our method learns the residual geometry solely on the source graph and requires no tuning on the target graph. Anomalies are detected via deviations in residual directions and distances to multiple centers, effectively avoiding inconsistencies in embedding distributions across domains.
>
> **Empirical evidence.**
> As shown in Table 1, under cross-domain zero-shot evaluation, Deep SVDD/SAD achieve only 45%–65% AUROC and often below 10% AUPRC on most target graphs, whereas our method outperforms them across seven domains. Deep SAD performs better on datasets with high anomaly separability, such as Weibo. However, our model performs better on graphs with strong anomaly inseparability while balancing weak anomaly separability. This shows that single-sphere contraction struggles with structural mismatches, while residual-direction geometry offers stronger robustness for cross-graph transfer.
>
> Table 1.Performance comparison with Deep SVDD and Deep SAD(AUROC/AUPRC).
> |Method|Facebook|ACM|Amazon|Cora|CiteSeer|Reddit|weibo|BlogCatalog|
> |-|-|-|-|-|-|-|-|-|
> |Deep SVDD|63.64/3.95|45.49/3.50|55.89/9.49|54.67/6.35|57.53/5.72|50.10/9.65|8.12/9.65|46.28/5.56|
> |Deep SAD|46.64/2.85|68.37/6.27|64.53/13.21|52.52/5.74|48.49/4.81|53.71/3.95|**94.56/83.20**|57.27/7.16|
> |Ours|**82.16/16.77**|**91.17/51.45**|**88.15/61.37**|**93.20/58.18**|**95.00/63.54**|**60.60/5.33**|93.31/70.06|**75.06/36.56**|
>
> **Q2: It is necessary to discuss the relevance to previous cross-domain works.**
>
> **A2:** Thank you for your valuable feedback. To better highlight the differences and advantages of our method compared to previous cross-domain anomaly detection works, we have provided a more detailed comparison:
>
> - COMMANDER[3] uses adversarial training and graph attention networks for domain adaptation, which aligns node representations between the source and target graphs, but still faces the issue of normal and anomalous feature overlap.
> - ACT[4] reduces the mixing of normal and anomalous representations through anomaly-aware contrastive alignment, but relies on target-domain data for alignment mechanisms.
> - CDFS-GAD[5] introduces a domain-adaptive graph contrastive learning and prompt tuning module to enhance cross-domain feature alignment. However, it requires fine-tuning on the downstream target domain.
> - Our method uses multi-scale residual-based HR and AR modules to detect both shared and unshared anomaly patterns, improving detection accuracy, especially when target domain data is unavailable. Unlike COMMANDER, ACT, and CDFS-GAD, which require target-domain data for training or fine-tuning, our method transfers learned anomaly patterns from the source graph without the need for fine-tuning.
>
> Below is a key comparison of the methods:
>
> Table 2: Comparison of Key Characteristics of Cross-Domain Graph Anomaly Detection Methods
> |Dimension|COMMANDER|ACT|CDFS-GAD|Ours (DR-GGAD)|
> |-|-|-|-|-|
> |Requires target-domain data for training|✔|✔|✔|✘|
> |Requires target-domain fine-tuning|✔|✔|✔|✘|
> |Explicitly handles Normal/Anomaly feature overlap|✘|✘|✘|✔|
> |Handles domain shifts without requiring target data|✘|✘|✘|✔|
>
> [1] Ruff et al., Deep SVDD, ICML'18
>
> [2] Ruff et al., Deep SAD, ICLR'20
>
> [3] Ding K et al. Cross-domain graph anomaly detection. TNNLS 2021.
>
> [4] Wang Q et al. Cross-domain graph anomaly detection via anomaly-aware contrastive alignment. AAAI 2023.
>
> [5] Chen J et al. Towards cross-domain few-shot graph anomaly detection. ICDM 2024.

---

> ### Author Response · Authors · 2025-11-17
>
> **We thank the reviewer for the valuable feedback and respond to points (3) and (4) below.**
> **Q3: Further details of the synergy between AR and HR in the BlogCatalog dataset**
>
> **A3:** We thank the reviewer for the valuable feedback. Regarding the limited synergy between $\mathcal{HRS}$ and $\mathcal{ARS}$ in the BlogCatalog dataset, we conducted further analysis:
>
> From the ablation experiment results (Table 3), it is clear that AUROC and AUPRC for the BlogCatalog dataset show significant improvements, with anomaly patterns driven by both feature and structural divergence. Specifically:
>
> - The ablation experiments show that both $\mathcal{HRS}$ and $\mathcal{ARS}$ effectively capture anomalies. $\mathcal{HRS}$ captures anomalies through feature shifts, while $\mathcal{ARS}$ captures them through structural residuals.
> - When $\lambda$ is between 0–0.7, performance fluctuations are small, suggesting that AR’s additional benefit is limited. This is mainly due to BlogCatalog's homophily of 0.8922, lower than other datasets like ACM (0.9282) and CiteSeer (0.9204). This heterogeneity restricts AR’s ability to capture local structural anomalies.
> - When $\lambda$ reaches 0.8, the combination of $\mathcal{HRS}$ and $\mathcal{ARS}$ results in a small performance boost, indicating that $\mathcal{HRS}$ further improves anomaly detection, especially for feature shift anomalies.
>
> Table 3. The ablation result on BlogCatalog
> |Method|AUROC|AUPRC|
> |-|-|-|
> |Backbone|60.29|9.20|
> |+HR|74.31|35.92|
> |+AR|74.43|36.05|
> |DR-GGAD|**75.06**|**36.56**|
>
> **Q4: How to identify the dominant abnormality factor on unseen graphs?**
>
> **A4:** We thank the reviewer for raising this important question. To clarify how
> $\lambda$ is determined for unseen graphs, we provide the complete selection and inheritance strategy below.
>
>  **(1) $\lambda$ Selection on Source Graphs**
>
> To determine how the dominant residual factor ($\mathcal{HRS}$ vs. $\mathcal{ARS}$) should be weighted on unseen graphs, we first perform systematic hyperparameter selection on the source domain datasets. On the four source graphs(PubMed,Flickr,Questions,YelpChi), we conduct a grid search over
> $\lambda \in \\{0,0.01,0.1,0.2,\dots,1.0\\}$
> and select the optimal value based on supervised validation accuracy during pretraining.
>
>  **(2) Two Clear Convergence Patterns**
>
> The source domain results exhibit two distinct convergence behaviors (see Table 4):
>
> - **PubMed (citation graph)->$\lambda = 0.1$**
>
>   Citation graphs typically have stable structures and regular neighborhoods. Structural anomalies—caused by inconsistent or incorrect local connectivity—are more effectively captured by AR, leading to a preference for smaller $\lambda$.
>
> - **Flickr / Questions / YelpChi (social / review graphs)->$\lambda = 0.9$**
>
>   Social and review graphs are highly noisy and structurally heterogeneous. Their irregularities often stem from local noise, making AR less effective. Anomalies in these graphs are more often reflected as feature deviations, so HR contributes more, leading to larger (\lambda).
>
> This divergence reflects inherent structural–feature differences across graph domains.
>
>  **(3) Cross Domain Inheritance**
>
> For unseen target graphs, we directly inherit the $\lambda$ value corresponding to their graph domain (citation / social / coreview), without tuning on the target domain:
>
> - Citation graphs->$\lambda = 0.1$
> - Social /Co-review graphs->$\lambda = 0.9$
>
> Prior studies[1] show that graphs from the same domain share similar statistical properties, such as structural noise levels, neighborhood consistency, and feature distribution, leading to similar dominant directions in the residual space (HR or AR dominant). Inheriting the domain-specific $\lambda$ helps preserve residual dominance learned from source graphs without tuning on target graphs.
>
> Similar domain-driven hyperparameters exist in prior work, like AnomalyGFM[2], which selects β based on global similarity. In contrast, our $\lambda$ is directly derived from source-domain residual dominance, providing more stable and interpretable cross-domain transfer.
>
>  **(4) Summary**
>
> The domain inherited $\lambda$ exploits consistent structural–semantic patterns within graph domains, enabling robust residual weighting on unseen graphs without target domain tuning.
>
> Table 4: Domain categories and the corresponding inherited $\lambda$ values.
> |Dataset|Citation(Injected)|Social(Injected)|Social(Real)|Co-review(Real)|$\lambda$|
> |-|-|-|-|-|-|
> |Cora|✔||||0.1|
> |CiteSeer|✔||||0.1|
> |ACM|✔||||0.1|
> |PubMed|✔(Pre-Train)||||0.1(optimal value)|
> |BlogCatalog||✔|||0.9|
> |Flickr||✔(Pre-Train)|||0.9(optimal value)|
> |Weibo|||✔||0.9|
> |Reddit|||✔||0.9|
> |Questions|||✔(Pre-Train)||0.9(optimal value)|
> |Amazon||||✔|0.9|
> |YelpChi||||✔(Pre-Train)|0.9(optimal value)|
>
> [1] Wang Y et al. Towards Graph Foundation Models: A Transferability Perspective.arXiv preprint.
>
> [2] Qiao H et al. AnomalyGFM: Graph foundation model for zero/few-shot anomaly detection. KDD2025.

---

> ### Author Response · Authors · 2025-11-17
>
> **We appreciate the reviewer's insightful comments. Our response to suggestion (5) is provided below.**
>
> **Q5: Why these GAD-Benchmark datasets were selected?**
>
> **A5:** We sincerely thank the reviewer for the valuable comments regarding our dataset selection strategy.
> Our dataset choices strictly follow the evaluation protocols commonly adopted in the field of generalized graph anomaly detection (GGAD). The goal is to maximize coverage across anomaly types, graph structures, feature dimensions, and domain distributions, thereby enabling a thorough evaluation of cross-domain generalization.
>
>  **1. Coverage of multiple real-world domains**
>
> Following the domain categorization of GAD-Benchmark[1] and the reviewer’s suggestion for broader domain coverage, we select datasets from **four representative real-world application scenarios**, including:
>
> - **Citation networks (with injected anomalies):** Cora, CiteSeer, ACM, PubMed
> - **Social networks (with injected or real anomalies):** BlogCatalog, Flickr, Facebook, Weibo, Reddit, Questions
> - **Co-review networks (with real anomalies):** Amazon, YelpChi
>
> This cross-domain mixture exposes the model to:
>
> - diverse node and edge connectivity patterns,
> - highly heterogeneous feature distributions,
> - and a combination of injected and real anomalies.
>
> Such diversity is crucial for constructing a general-purpose GAD model that can truly generalize to **unseen graphs**.
>
>  **2. Diversity of anomaly types and difficulty levels**
> Our dataset suite incorporates:
>
> - **Injected anomalies** (structural/attribute corruption with high controllability),
> - **Real anomalies** (fraud, bots, abnormal users with higher complexity),
> - **A wide range of anomaly ratios (2–10%)**,
> - **Substantially different AnD (Anomaly non-Discriminativity) levels**, covering “easy → medium → hard” difficulty tiers.
>
> This design ensures that the model does not overfit to a single anomaly pattern but learns **cross-anomaly residual patterns**, which is the central objective of GGAD.
>
> Appendix A further provides the AnD values of all datasets, demonstrating the broad difficulty spectrum.
>
>  **3. Consistency with widely adopted GGAD evaluation protocols**
>
> Our dataset selection strictly adheres to **GAD-Benchmark**, as well as recent representative GGAD works such as
> **ARC[2]**:
>
> - In each category, **the largest graph is used as the training set**,
> - The remaining datasets serve as **zero-shot target graphs**.
>
> Following this standardized evaluation protocol ensures reproducibility and fair comparison with existing literature.
>
>  **4. Additional large-scale GAD-Benchmark datasets for enhanced validation**
>
> To further broaden the evaluation scope, we additionally include **three real-world datasets from GAD-Benchmark**:
>
> **Table 1. Detailed statistics of the additional datasets.**
>
> |Name|Nodes|Edges|Dim|Anomaly|Anomaly Type|Size|Domain|
> |--|--|--|--|--|--|--|--|
> |T-Finance|39,357|21,222,543|10|4.60%|Real|Large|Finance network|
> |CS|6,805|18,333|8|22.69%|Real|Small|Co-purchase|
> |Photo|7,535|119,043|745|9.2%|Real|Small|Co-purchase|
>
> These datasets differ significantly from our primary suite in terms of domain (finance, co-purchase), structural density (T-Finance is extremely dense), and feature dimensionality (8–745).
> Yet our method consistently achieves strong performance:
>
>  **Table 2. Additional results on GAD-Benchmark datasets.**
>
> |Method|T-Finance(AUROC)|CS(AUROC)|Photo(AUROC)|
> |--|--|--|--|
> |DOMINANT(2019)|OOM|0.6214|0.5010|
> |CoLA(2021)|0.5134|0.7241|0.5618|
> |TAM(2023)|OOM|0.6995|0.5724|
> |GCTAM(2025)|OOM|0.7133|0.6042|
> |SmoothGNN(2025)|0.3979|0.4520|0.4791|
> |UNPrompt(2024)|0.2386|0.7108|0.5193|
> |AnomalyGFM(2025)|0.6757|0.5676|0.5157|
> |ARC(2024)|0.6410|0.8273|0.7480|
> |**DR-GGAD**|**0.7558**|**0.9801**|**0.7507**|
>
> These results demonstrate that **DR-GGAD maintains excellent generalization capabilities even on domains, densities, and feature spaces entirely different from those seen during training**.
>
> [1] Tang J et al. GADBench: Revisiting and benchmarking supervised graph anomaly detection. NeurIPS 2023.
>
> [2] Liu Y et al. ARC: A generalist graph anomaly detector with in-context learning. NeurIPS 2024.

---

> ### Author Response · Authors · 2025-11-26
>
> Dear Reviewer,
>
> We hope this message finds you well. First, we would like to express our sincere gratitude for taking the time to carefully review our work and for providing frank and insightful comments. Your feedback is extremely valuable in helping us better understand and improve our manuscript.
>
> In response to your review, we have made a series of substantial revisions and additions during the current discussion phase, and we have provided detailed clarifications and new experimental results above. We sincerely hope that these additional efforts have, at least to some extent, addressed your main concerns regarding the methodology of this work.
>
> As the discussion phase is drawing to a close, if any parts of our responses remain insufficient or unclear, we would be very grateful if you could continue to point them out. We will do our utmost to further refine and supplement the manuscript within the remaining time of the discussion.
>
> Once again, we sincerely thank you for the time and effort you have devoted to our submission, as well as for your rigorous evaluation and constructive suggestions on our work.
>
> Sincerely,
>
> The Authors

---

### Official Review · Reviewer_ozsE · 2025-11-01

**Soundness:** 2
**Presentation:** 2
**Contribution:** 2
**Rating:** 4
**Confidence:** 4

**Summary:**

The paper introduces $\mathcal{A}n\mathcal{D}$, a graph anomaly detection framework that leverages multi-scale residual representations derived from successive graph neural network layers to detect anomalous nodes and substructures. It constructs domain-invariant reference centers and measures affinity residuals between node embeddings and these references to identify deviations indicative of anomalies. The method aims to capture both feature-level and topological irregularities in graphs without relying on labeled anomalies or reconstruction losses.

**Strengths:**

1.	The paper is easy to follow.
2.	The performance seems improved a lot.
3.	The statistical results are comprehensive.

**Weaknesses:**

1.	Is the dimensionality of the visualized embeddings in Figure 1 directly set to 2? Since t-SNE projects data into a 2D space, the resulting distribution differs from the actual data distribution.
2.	The authors should provide a discussion comparing $\mathcal{A}n\mathcal{D}$ with existing distribution-level metrics such as MMD. While MMD measures distributional distances, what advantages does $\mathcal{A}n\mathcal{D}$ offer?
3.	The authors should clarify the technical differences between the proposed method and clustering-based approaches.
4.	The paper claims that multi-scale residuals from successive GNN layers form a domain-invariant reference, but no theoretical justification or citation is provided to support this statement.
5.	The authors state that the Affinity Residual component can reveal anomalies primarily hidden in the topology. However, no experimental evidence supports this claim. It is also recommended to compare with subgraph anomaly detection methods to substantiate this advantage.
6.	The authors mention that no existing methods explicitly address representation overlap. However, representation overlap appears related to clustering operations that aim to increase inter-class separation, such as works [1], [2]. Please discuss how the proposed method differs from or improves upon those clustering-related works.
7.	The paper lacks one-class classification baselines such as OCSVM. The authors claim their method remains effective across various unseen graphs. If the decision boundary tightly encloses the normal data, one-class models may also generalize well. Please include a comparison or discussion on this point.
8.	An ablation study comparing the concatenation of residuals from all layers versus using only the final-layer residual is missing. The authors should explain why concatenation was chosen instead of alternative fusion strategies (e.g., addition or averaging).
9.	It remains unclear how the proposed method extracts semantically meaningful embeddings without the guidance of a reconstruction or logits-based loss.
10.	Why was K-Means chosen for center initialization instead of Spectral Clustering? Please include comparisons and also provide results using K-Means directly for anomaly detection.
11.	The problem definition is unclear. Does the center construction step rely solely on normal data?
12.	What is the meaning of $\epsilon$, and what value is used in the experiments?
13.	More visual comparisons for the ablation studies should be presented to illustrate their effects intuitively.
14.	How does the proposed method handle source data with inconsistent feature dimensions?
15.	The dataset name in Figure 3 is inconsistent with that in the experimental table; please correct it.
16.	Please analyze why the proposed method achieves lower AUPRC on the Reddit and Weibo datasets.
17.	The manuscript contains multiple formatting issues. Please proofread carefully for consistency and correctness.

**Reference:**

 [1] "discDC: Unsupervised discriminative deep image clustering via confidence-driven self-labeling." Pattern Recognition (2025): 112382.

[2] "Towards feature distribution alignment and diversity enhancement for data-free quantization." 2022 IEEE International Conference on Data Mining (ICDM). IEEE, 2022.

**Questions:**

Please refer to Weaknesses.

---

> ### Author Response · Authors · 2025-11-19
>
> **We appreciate Reviewer ozsE for the positive review and constructive comments. Our point-by-point responses to points (1) and (1) are provided below:**
>
> **Q1: t-SNE 2D projection causes distribution differences.**
>
> **A1:** Thank you very much for your valuable suggestions.
>
> We adopted a 2D t-SNE projection to provide an intuitive visualization of normal and anomalous node distributions. To ensure that our motivation is not affected by the dimensionality reduction of t-SNE, and to more accurately characterize the  Anomaly non-Discriminativity($\mathcal{A}n\mathcal{D}$) issue, we further validated the consistency between the visualizations and the actual high-dimensional distribution using the $\mathcal{A}n\mathcal{D}^{\ast}$ score.
>
> Specifically, we computed the $\mathcal{A}n\mathcal{D}^{\ast}$ scores for each dataset and examined their correlation with AUROC (Table 1). The results show a strong relationship, with a high Pearson correlation of 0.7508. Higher $\mathcal{A}n\mathcal{D}^{\ast}$ scores are associated with lower AUROC values, indicating that greater overlap between normal and anomaly nodes corresponds to increased detection difficulty.
>
> According to the definitions provided in Equations (1)–(2) of our paper, the $\mathcal{A}n\mathcal{D}^{\ast}$ score quantifies the degree of distance overlap between normal and anomalous nodes. A higher $\mathcal{A}n\mathcal{D}^{\ast}$ value reflects stronger inter-class entanglement and weaker separability in the embedding space, thereby making anomaly detection inherently more challenging.
>
> This result further demonstrates that the distribution issues are independent of the t-SNE projection. t-SNE is only used for intuitive visualization, and the core conclusions remain consistent under the $\mathcal{A}n\mathcal{D}^{\ast}$ metric.
>
> Table 1 $\mathcal{A}n\mathcal{D}^{\ast}$ and AUROC for different datasets.
> |Dataset|Questions|YelpChi|PubMed|Flickr|Cora|CiteSeer|ACM|BlogCatalog|Facebook|Weibo|Reddit|Amazon|
> |--|--|--|--|--|--|--|--|--|--|--|--|--|
> |$\mathcal{A}n\mathcal{D}^{\ast}$|0.4079|0.4197|0.4061|0.4397|0.5190|0.5294|0.4159|0.4408|0.5272|0.2745|0.5964|0.5163|
> |AUROC|67.90|68.98|54.82|58.25|59.64|60.27|60.49|58.91|55.68|70.56|42.02|45.49|
>
> **Q2: What distinguishes your method from MMD in measuring distributional differences?**
>
> **A2:** We appreciate the reviewer’s valuable comments. Below is our discussion comparing Anomaly non-Discriminativity ($\mathcal{A}n\mathcal{D}$) with Maximum Mean Discrepancy (MMD).
>
> **(1) Conceptual distinction between AnD and MMD**:
> AnD measures the **class-conditional overlap** between normal and anomalous nodes, capturing the loss of separability within a single embedding space. In contrast, MMD is a general **marginal distribution discrepancy** metric that quantifies the shift between two sets of embeddings, applicable to both **inter-domain differences** (e.g., ($P_s(Z)$) vs. ($P_t(Z)$)) and **intra-domain differences** (normal vs. anomaly nodes within the same graph). The two metrics thus focus on fundamentally different aspects.
> **(2) Empirical comparison AnD shows the strongest correlation with AUROC**:
> We compute Inter-domain MMD, Intra-domain MMD, and $\mathcal{A}n\mathcal{D}^{\ast}$ across eight target datasets and report their Pearson correlations with AUROC (Table 2).
>
> - Inter-domain MMD -0.7246
> - Intra-domain MMD -0.2302
> - $\mathcal{A}n\mathcal{D}^{\ast}$ 0.7508 (highest)
> $\mathcal{A}n\mathcal{D}^{\ast}$ exhibits the strongest and most stable association with detection performance.
>
> **(3) Case study Amazon**
> On Amazon, Intra-domain MMD is relatively high while AUROC is low. however,$\mathcal{A}n\mathcal{D}^{\ast}$ reaches 0.5163, directly reflecting the severe overlap between normal and anomalous nodes and aligning closely with the observed performance drop. This highlights AnD's superior explanatory power.
>
> **(4) Summary Why MMD correlates weakly with AUROC**
> MMD describes global distribution shift, which may not directly affect class separability two distributions can be far apart yet remain well separated by class, or vice versa. Hence, its relation to AUROC is inherently indirect and confounded by multiple factors.
> In contrast, AnD directly quantifies class-overlap (discriminability degradation), making it more strongly and consistently aligned with anomaly detection difficulty and performance. We have further elaborated on this relationship in the appendix of the revised manuscript.
>
> Table 2 Comparison of Distribution-Shift Metrics and the Correlation with AUROC
>
> |Dataset|Cora|CiteSeer|ACM|BlogCatalog|Facebook|Weibo|Reddit|Amazon|Pearson correlation coefficient|
> |--|--|--|--|--|--|--|--|--|--|
> |Inter-domain MMD|0.8545|1.0002|0.5933|0.5421|0.8077|0.9478|0.5189|0.3416|-0.7246|
> |Intra-domain MMD|0.0115|0.0092|0.0454|0.1911|0.0382|0.01362|0.0002|0.1758|-0.2302|
> |AnD|0.5190|0.5294|0.4159|0.4408|0.5272|0.2745|0.5964|0.5163|0.7508|
> |AUROC|59.64|60.27|60.49|58.91|55.68|70.56|42.02|45.49|-|

---

> ### Author Response · Authors · 2025-11-19
>
> **We appreciate the reviewer’s valuable feedback. Below is our detailed response to suggestion (3).**
>
> **Q3: Clarify methodological distinctions from clustering-based approaches.**
>
> A3: Thank you to the reviewers for their valuable feedback. We would like to further clarify the differences between our method (DR-GGAD) and clustering methods such as K-means and CARE[1], and provide a comparative analysis with experimental results.
>
> **1. Clarification of Technical Differences:**
>
> - **DR-GGAD**: DR-GGAD learns the "normal hyper-residual centers" (HR) and "neighborhood directional consistency" (AR) constraints from the source domain and performs zero-shot scoring in the target domain without requiring re-clustering. Unlike traditional clustering methods, DR-GGAD does not rely on re-clustering the target graph but instead introduces distance and consistency constraints during training to more accurately model anomalies, avoiding the local clustering assumption problem inherent in traditional clustering methods.
> - **CARE**: CARE introduces a clustering-aware method by enhancing the adjacency matrix with soft labels, combining multi-view information and contrastive learning to flexibly capture node similarity and anomalies. By simulating clustering structures with soft labels, it avoids the fixed partitioning used in traditional clustering methods.
>
> **2. Comparison between DR-GGAD and Clustering-based methods:**
>
> Our experimental comparisons with clustering methods show that DR-GGAD outperforms CARE on most datasets.
>
> |Method|Facebook|ACM|Amazon|Cora|CiteSeer|Reddit|weibo|BlogCatalog|
> |--|--|--|--|--|--|--|--|--|
> |K-means|50.06/2.78|72.77/30.59|57.75/7.45|57.71/16.77|55.44/15.28|41.85/2.63|93.90/77.05|72.81/32.21|
> |CARE|76.39/6.00|67.43/18.95|70.24/17.85|30.25/3.65|33.91/3.23|45.64/3.25|**97.15/87.12**|66.72/32.83|
> |DR-GGAD|**82.16/16.77**|**91.17/51.45**|**88.15/61.37**|**93.20/58.18**|**95.00/63.54**|**60.60/5.33**|93.31/70.06|**75.06/36.56**|
>
> **Experimental Analysis:**
>
> From the experimental results, **DR-GGAD** performs excellently on multiple datasets, especially on **Amazon** and **CiteSeer**, where it significantly outperforms **CARE**. The analysis is as follows:
>
> - On the **Amazon** dataset, DR-GGAD achieves an AUROC of **88.15**, which is significantly higher than CARE’s **70.24**. This indicates that DR-GGAD is better at capturing feature anomalies in the graph, especially when node relationships are heterogeneous, showing more stable performance.
> - On the **Weibo** dataset, **CARE** has an AUROC of **97.15**, which is higher than DR-GGAD’s **93.31**. DR-GGAD's performance on this dataset is slightly lower. **CARE** performs better by optimizing node affinities through contrastive learning, allowing it to capture local structures more effectively. However, DR-GGAD models both global and local structural anomalies with AR and HR, maintaining an advantage in complex graphs.
>
> **3. Summary:**
>
> - **DR-GGAD** performs excellently across multiple datasets, particularly on **Amazon** and **CiteSeer**, where it accurately models complex graph structures and captures the intricate relationships between nodes.
> - **CARE** performs well on the **weibo** dataset, benefiting from its optimization of local node affinities. However, DR-GGAD's performance remains more stable and comprehensive on more complex graph datasets.
> - We also compared **DR-GGAD** with **K-Means**, demonstrating that DR-GGAD is more advantageous in handling structural anomalies in complex graphs than traditional clustering methods.
>
> **Conclusion:**
>
> Through the above analysis, we compared the technical differences between DR-GGAD, K-Means, and CARE. DR-GGAD is more efficient in handling structurally complex graph data and has demonstrated superior performance across multiple datasets, especially on graphs with significant feature deviation anomalies.
>
> We have added relevant discussions in the appendix of the revised manuscript to enhance the explanation and clarity of the differences between our method and clustering approaches.
>
> [1] Zheng L et al. Cluster aware graph anomaly detection. WWW 2025.

---

> ### Author Response · Authors · 2025-11-19
>
> **We sincerely thank the reviewer for the valuable suggestions. Our detailed responses to points (4), (11), and (12) are presented below.**
>
> **Q4: Lacks justification for the domain-invariance claim of multi-scale residuals.**
>
> **A4:** We appreciate the reviewer’s suggestions. We further clarify the design rationale behind the multi-scale residual mechanism and provide additional related research and references.
> Our method is based on the following two design principles:
>
> - **Residual = Neighborhood Aggregation - Ego node:** This operation highlights the difference between the node and its neighbors, emphasizing not just the node's own semantics but also how it deviates from its neighbors. This difference is a manifestation of local affinity, whether a node’s features or structure are consistent with or deviate from those of its neighbors[1]. Research has shown that anomalous deviations in local affinity are key indicators in graph anomaly detection and exhibit strong generalization across different graph domains[1],[2],[3].
> - **High-Pass/Differential Filter Perspective:** The residual operation can also be viewed as a high-pass filter, which attenuates the strong homophilic signals commonly present among neighbors (a typical characteristic of normal nodes) while retaining or enhancing those "high-frequency" or heterophilic (structural deviation) signals typically associated with anomalies. For instance, studies have observed that anomalous nodes cause a "right-shift" in the graph signal spectrum, with energy in the spectrum shifting from low to high frequencies[4],[5].
>
> Building on these two points, the multi-scale residual mechanism (i.e., calculating the residual through successive GNN layers or multi-hop neighborhood aggregation) allows the model to capture not only the 1-hop neighborhood differences but also higher-order structural or feature deviations. This mechanism facilitates the extraction of domain-invariant representations across different graph domains[2]. In other words, although source and target domains may differ in feature dimensions, structural distributions, and neighborhood patterns, the distribution of "node-neighbor residuals" exhibits stronger transferability, making it a reliable reference. Based on this logic, DR-GGAD constructs a transferable and relatively domain-invariant reference feature space through multi-scale residuals. Additionally, we have included further clarifications in the appendix of the revised manuscript.
>
> [1] Qiao H et al. Truncated affinity maximization: One-class homophily modeling for graph anomaly detection. NeurIPS 2023.
>
> [2] Liu Y et al. ARC: A generalist graph anomaly detector with in-context learning. NeurIPS 2024.
>
> [3] Qiao H et al. Deep graph anomaly detection: A survey and new perspectives. TKDE 2025.
>
> [4] Tang J et al. Rethinking graph neural networks for anomaly detection. ICML 2022.
>
> [5] Li S et al. High-pass graph convolutional network for enhanced anomaly detection: A novel approach. arXiv 2024.
>
> **Q11: The problem definition is unclear. Does the center construction step rely solely on normal data?**
>
> **A11:** Thank you for your insightful feedback. To clarify, the proposed Hyper Residual (HR) centers are learned exclusively from normal nodes in the labeled source-domain graphs. During center construction (Eq. 8), only normal node residuals contribute to the clustering process. Abnormal nodes are not involved in estimating the centers; they only appear in the repulsion term of the loss to enforce a margin-based separation.
>
> Importantly, the target-domain graphs are entirely unlabeled. No labels, re-training, or center updates are performed during inference. All centers and parameters are fixed, ensuring a strict zero-shot GGAD setting. We have clarified this point explicitly in the revised manuscript to enhance the precision and transparency of our problem definition.
>
> **Q12: What is the meaning of $\epsilon=1$, and what value is used in the experiments?**
>
> **A12:** We thank the reviewer for the constructive suggestion. In our model, the margin ($\epsilon=1$) controls the minimum separation required between the multi-scale normal residual centers and the residual representations of abnormal nodes. We encourage the residuals of normal nodes to cluster tightly around the multi-scale normal residual centers, while ensuring that the residuals of abnormal nodes remain at least ($\epsilon=1$) away from this center set. We have incorporated the corresponding clarification in the revised manuscript to improve the readability and clarity of the paper.

---

> ### Author Response · Authors · 2025-11-19
>
> **We sincerely thank the reviewer for their constructive suggestions. Below we provide a detailed response to point (5).**
>
> **Q5: No evidence supports that Affinity Residual reveals topology-based anomalies. Compare with subgraph detection methods to validate this.**
>
> **A5:** We would like to thank the reviewer for the valuable suggestion. We provide comprehensive justification from both the design motivation and new experimental perspectives regarding the AR module.
>
> **(1) Design Motivation**
>
> AR (Affinity Residual) measures the consistency of node residuals with the neighborhood. Normal nodes maintain stable residual directional consistency due to regular neighborhood aggregation, while structural anomalies cause abrupt changes, leading to significant residual directional mismatch. Unlike detection methods that rely on absolute features, AR primarily focuses on:
>
> - Consistency of neighborhood structure;
> - Consistency of residual direction
> - Topological deviation between neighbors
>
> Therefore, AR is inherently sensitive to structural anomalies, such as: Motif anomalies, Anomalous cross-community edges, Connections that deviate from homophily. Based on this design motivation, we conducted two types of verification experiments.
>
> **(2) Experiment I: Structural Anomaly Injection**
>
> We followed the anomaly injection protocol in previous works[1],[2] and conducted structural perturbation experiments on multiple datasets. Specifically, we injected fully connected small cliques of size $p$ into some anomalous nodes to simulate structure-dominant anomalies (the larger $p$ is, the stronger the structural anomaly injection). The experimental results (AUROC) are as follows:
>
> |Method|Facebook|ACM|Cora|CiteSeer|Reddit|Weibo|BlogCatalog|Amazon|
> |--|--|--|--|--|--|--|--|--|
> |AR-Only|82.16|90.85|93.20|94.03|58.04|88.35|74.43|61.79|
> |AR+Structural noise(p=2)|82.48|92.17|94.63|95.19|61.77|91.21|74.60|59.03|
> |AR+Structural noise(p=4)|83.28|93.88|94.58|95.93|68.10|92.47|74.76|57.33|
> |AR+Structural noise(p=6)|83.62|94.11|95.32|96.51|69.84|92.86|75.42|57.15|
>
> **Experimental Analysis**
>
> - As the structural anomaly increases ($p$ from 2->6), AR's AUROC significantly improves. This indicates that the stronger the structural anomaly, the more easily AR detects it.
> - The improvement is most significant on typical homogeneous graphs like Cora, CiteSeer, and ACM. This shows that AR is particularly sensitive to topological perturbations.
> - Even on datasets with stronger noise, such as Reddit and Weibo, AR still shows stable improvement.
>
> Hence, we conclude that AR responds monotonically to structural perturbations and robustly captures structural anomalies.
>
> **(3) Experiment II: Comparison with Subgraph Anomaly Detection Methods**
>
> To validate AR's advantages for "structure-dominant anomalies," we compare it with two representative subgraph anomaly detection methods: **GAD-EBM[3]** and **GCAD[4]**.
>
> - GAD-EBM detects anomalies by comparing the likelihood of a subgraph with its neighbors using score matching.
> - GCAD enhances topological anomaly detection by centralizing and leveraging the structural features of subgraphs.
>
> Both methods explicitly utilize subgraph structures, making them effective benchmarks for evaluating AR's ability to capture structural anomalies.
>
> The results (AUROC) are as follows:
>
> |Method|Facebook|ACM|Amazon|Cora|CiteSeer|Reddit|Weibo|BlogCatalog|
> |--|--|--|--|--|--|--|--|--|
> |GADEBM|32.87|58.80|50.65|59.38|60.17|53.28|47.10|53.00|
> |GCAD|50.64|79.43|57.89|50.00|50.39|53.51|48.79|64.94|
> |Our|82.16|91.17|88.15|93.20|95.00|60.60|93.31|75.06|
>
> **Experimental Analysis**
>
> - AR outperforms both subgraph baselines across all 8 datasets.
> - AR's advantage is particularly prominent on structure-dominant datasets (such as Cora, CiteSeer, and ACM).
> - In cross-domain settings, subgraph methods rely on specific subgraph patterns, whereas AR is based on “residual directional consistency” and does not depend on specific motifs, thus exhibiting stronger generalization capabilities.
>
> **(4) Conclusion**
>
> Combining motivation and experimental results, we conclude the following:
>
> - AR is sensitive to the disruption of local structural consistency, performing better as structural anomalies increase, particularly excelling in structure-dominant anomalies.
> - Compared to subgraph methods, AR maintains its advantage in cross-domain settings by learning residual direction distributions rather than relying on fixed subgraph templates.
>
> [1] Ding K et al. Deep anomaly detection on attributed networks. SDM 2019.
>
> [2] Liu Y et al. Anomaly detection on attributed networks via contrastive self-supervised learning. TNNLS 2021.
>
> [3] Roy A et al. GAD-EBM: Graph anomaly detection using energy-based models. NeurIPS Workshop 2023.
>
> [4] Zhuang Z et al. Subgraph centralization: A necessary step for graph anomaly detection. SDM 2023.

---

> ### Author Response · Authors · 2025-11-19
>
> **We appreciate the reviewer's valuable comments and provide the following response to suggestion (6).**
>
> **Q6: How does your method differ from clustering-based representation separation approaches?**
>
> **A6:** Thank you for your valuable suggestion, which has helped us clarify the conceptual distinction between our approach and clustering-based representation separation methods.
>
> **(1) discDC[1] and ClusterQ[2] essentially follow a target-domain clustering redistribution paradigm:**
> These methods reshape feature separability by re-clustering **within the target domain**. ClusterQ clusters BN statistics, while discDC continuously updates cluster structures via self-labeling to optimize intra-cluster compactness and inter-cluster separation. Consequently, their cluster structures, centers, and boundaries all shift with the target-domain distribution.
>
> **(2) Our method is fundamentally different, built on a *source-domain hard constraint* rather than target-domain clustering:**
> We do not perform clustering or center updating in the target domain. In anomaly detection, node representations often exhibit camouflage and semantic mixing, making direct clustering unreliable. To address this, we first obtain more discriminative residual features through multi-scale residual extraction; we then learn a **source-domain residual center and directional prior**, which is iteratively refined and solidified into a stable, generalizable, and target-agnostic constraint.
>
> **(3) In the target domain, we apply zero-adaptation deviation scoring:**
> Target samples are never clustered; instead, they are evaluated solely based on their deviation from the residual center, effectively suppressing directional overlap and avoiding the overhead of target-domain clustering. Therefore, our method forms a **residual-space constraint framework** grounded in source-domain priors.
>
> **(4) Comparison with the target-domain k-means baseline:**
> As shown in Table 6, directly running k-means in the target domain results in consistently poor AUROC/AUPRC, indicating that while raw embeddings have some discriminatory power, they have limitations in effectively distinguishing normal from anomalous nodes. In contrast, our multi-scale residual extraction and source-domain residual center provide a more discriminative scoring reference, yielding substantially better performance across most datasets.
>
> To ensure completeness, we have supplemented the appendix G.2 with further discussion and detailed explanations.
>
> Table 6. Performance Comparison with Target-Domain k-means Baseline(AUROC/AUPRC)
>
> |Method|Facebook|ACM|Amazon|Cora|CiteSeer|Reddit|Weibo|BlogCatalog|
> |--|--|--|--|--|--|--|--|--|
> |K-means|50.06|72.77|57.75|57.71|55.44|41.85|**93.90**|72.81|
> |Ours|**82.16**|**91.17**|**88.15**|**93.20**|**95.00**|**60.60**|93.31|**75.06**|
>
> [1] Cai J et al. discDC: Unsupervised discriminative deep image clustering via confidence-driven self-labeling. Pattern Recognition 2025.
>
> [2] Gao Y et al. Towards feature distribution alignment and diversity enhancement for data-free quantization. ICDM 2022.

---

> ### Author Response · Authors · 2025-11-19
>
> **We appreciate the reviewer's valuable comments and provide the following responses to suggestions (7),(8).**
>
> **Q7: Please compare with one-class methods like OCSVM.**
>
> **A7**: Thank you for the valuable suggestion. We have added representative one-class baselines, including **OCSVM**[1], **Deep SVDD**[2], and **Deep SAD**[3], and evaluated them under the same protocol (Table 7).
>
> **(1) Shared Boundary Mechanism**:
> These methods learn compact boundaries around normal data (e.g., hyperspheres or one-class functions), assuming anomalies lie far from normal clusters. OCSVM maximizes the one-class margin, Deep SVDD compresses normal nodes into a minimal hypersphere, and Deep SAD incorporates limited anomaly supervision. However, in graphs, anomalous nodes often exhibit representation camouflage and semantic mixing, making embedding-based boundaries unreliable.
>
> **(2) Our Residual-Prior Approach**:
> Our method extracts multi-scale residuals to capture anomaly-specific deviations—addressing one-class methods' limitations. By learning a domain-agnostic residual center and directional prior from the source graph, we enable direct anomaly scoring in target domains without optimization. Experiments across seven datasets show consistent and substantial gains over one-class baselines (e.g., +18.52 AUROC on Facebook), with competitive results also achieved on the Weibo dataset, demonstrating strong robustness and transferability.
>
> In the revised version of the manuscript, we have added additional discussions in the Related Work section, further enriching the content on relevant methodologies.
>
> Table 7 Comparison with One-Class Anomaly Detection Baselines (AUROC)
> |Method|Facebook|ACM|Amazon|Cora|CiteSeer|Reddit|Weibo|BlogCatalog|
> |--|--|--|--|--|--|--|--|--|
> |OCSVM|43.44|25.13|30.83|25.71|22.80|48.48|22.28|24.62|
> |Deep SVDD|63.64|45.49|55.89|54.67|57.53|50.10|12.56|46.28|
> |Deep SAD|46.64|68.37|64.53|52.52|48.49|53.71|**94.56**|57.27|
> |**Our**|**82.16**|**91.17**|**88.15**|**93.20**|**95.00**|**60.60**|93.31|**75.06**|
>
> **Q8: Please explain the choice of concatenated multi-layer residuals over other fusion strategies.**
>
> **A8:** We appreciate the reviewer’s suggestion to include an ablation study comparing concatenated multi-layer residuals with other fusion strategies. Below, we further clarify the motivation and summarize the experimental findings.
>
> **(1) Motivation: Preserving Multi-Scale Deviations and Layer-Wise Semantic Frequencies**
>
> Residuals from different GNN layers encode distinct semantic frequencies:
>
> - shallow layers capture **local structural deviations**,
> - deeper layers reflect **global semantic drift**.
>
> Simple fusion strategies (e.g., summation or averaging) compress these heterogeneous signals into a single vector, smoothing or eliminating crucial cross-layer differences and losing fine-grained cues needed for anomaly detection. In contrast, **concatenation preserves the full multi-scale residual spectrum**, enabling the model to exploit both local perturbations and high-level semantic variations for a more discriminative anomaly space.
>
> **(2) Ablation Results: Concatenation Achieves the Best Performance**
>
> Table 8 compares **Sum**, **Mean**, **Max/Min**, **Final-layer-only**, and **Concatenation (DR-GGAD)**, and results consistently show:
>
> - summation, averaging, and max/min pooling discard key cross-layer residual information;
> - using only the final layer ignores shallow-layer deviations;
> - concatenation achieves the best or near-best performance on 7–8 datasets, with clear gains on Amazon, Reddit, and BlogCatalog where multi-scale anomalies emerge.
>
> These findings confirm that multi-layer residuals provide complementary structural and semantic deviation signals. Concatenation preserves these cross-scale cues and substantially improves cross-graph generalization and anomaly separability.
>
> We have also added further discussion of these observations in the appendix of the revised manuscript.
>
> Table 8. Ablation Study of Multi-Layer Residual Fusion Strategies (AUROC/AUPRC).
>
> |Method|Facebook|ACM|Amazon|Cora|CiteSeer|Reddit|Weibo|BlogCatalog|
> |--|--|--|--|--|--|--|--|--|
> |**Sum**|81.50/16.56|91.01/51.25|87.43/56.05|92.90/57.55|**95.11/64.31**|58.20/4.86|93.05/69.43|74.22/35.50|
> |**Mean**|81.95/16.15|91.17/51.40|85.02/40.05|93.04/55.54|94.56/61.63|58.49/4.87|93.08/68.66|74.25/37.58|
> |**Max**|82.00/16.40|91.05/51.41|85.54/43.54|93.05/55.72|94.61/62.34|58.48/4.87|92.85/69.43|74.32/35.47|
> |**Min**|81.60/16.41|91.10/51.35|85.84/44.63|92.82/55.45|94.61/62.30|58.78/4.80|93.05/69.44|74.17/36.47|
> |**Final-layer-only**|**82.51**/16.75|91.00/51.20|85.42/42.60|93.13/56.13|94.56/62.85|58.56/4.80|93.10/69.64|74.14/35.98|
> |**Concatenation**|82.16/**16.77**|**91.17 / 51.45**|**88.15 / 61.37**|**93.20 /58.18**|95.00/63.54|**60.60 / 5.33**|**93.31 / 70.06**|**75.06 / 36.56**|
>
> References:
>
> [1] Schölkopf et al., OCSVM, Neural Computation, 2001
>
> [2] Ruff et al., Deep SVDD, ICML'18
>
> [3] Ruff et al., Deep SAD, ICLR'20

---

> ### Author Response · Authors · 2025-11-19
>
> **Q9: How does the method learn meaningful semantics without reconstruction or classification loss?**
>
> **A9:** We sincerely appreciate the reviewer’s interest in how our model obtains meaningful semantic representations. Below we clarify the semantic learning mechanism of DR-GGAD.
>
> **(1)Label Supervision**:
> Although DR-GGAD does not use an explicit classification head, it directly incorporates normal/abnormal labels in the source domain through a center–residual contrastive loss. Normal nodes are pulled toward a shared residual center, while abnormal nodes are pushed away, forming a clear and learnable semantic margin that defines a stable “normal center.”
>
> **(2)Structural Prior**:
> The AR module introduces local structural semantics: nodes whose neighborhoods exhibit regular structural patterns align in residual direction, whereas nodes with structural irregularities or semantic inconsistencies deviate. This neighborhood-consistency constraint allows DR-GGAD to capture meaningful local structural cues without reconstruction-based supervision.
>
> **(3)Global + Local Semantics**:
> HR provides a cross-domain shared global residual center, ensuring stable normal semantics across different graphs. AR contributes local structural coherence. Together, these two mechanisms create an effective semantic supervision pathway, enabling residual embeddings to maintain both global separability and sensitivity to local structural deviations.
>
> **(4)Target-Domain Mapping**
> During inference, target-domain residuals are projected into the source-domain residual space, where the learned normal residual center serves as a fixed semantic reference. Residuals near the center indicate consistency with normal patterns, while directional or magnitude deviations reflect violations of the learned normal prior. Because the residual center and directional prior are already stabilized during source training, target residuals naturally fall into a **cross-domain consistent and interpretable semantic coordinate system**, supporting reliable anomaly detection on unseen graphs.
>
> We have supplemented the Methodology sections in the revised manuscript with further elaboration on the design rationale to ensure clarity and interpretability of the methodological design.

---

> ### Author Response · Authors · 2025-11-19
>
> **Q10: On the Choice of K-Means vs. Spectral Clustering & K-Means Baseline**
>
> **A10:** We thank the reviewer for raising the question regarding our use of K-Means versus spectral clustering. As detailed below, we compare the two methods in terms of effectiveness, modeling requirements, and computational feasibility, and we additionally report a K-Means–only baseline to complete the analysis.
>
> **(1)Method Difference**:
> We evaluated both K-Means and spectral clustering for constructing the Hyper Residual centers. K-Means operates directly in the residual feature space by iteratively minimizing the Euclidean distance to cluster centroids. In contrast, spectral clustering requires building the graph Laplacian, performing eigen-decomposition on its top-k eigenvectors, and then running K-Means in the resulting spectral embedding space. These approaches respectively represent distance-driven and structure-driven clustering paradigms.
>
> **(2)Experimental Results**:
> We tested three configurations: a simple K-Means baseline, a variant replacing K-Means with spectral clustering inside the HR module, and the full method. While the standalone K-Means baseline performs noticeably worse, the HR/AR framework makes the K-Means–HR and spectral–HR variants nearly indistinguishable—performance differences remain below 0.3% (e.g., Cora: 93.20 vs. 93.14, ACM: 91.17 vs. 91.11, Amazon: 88.15 vs. 88.14). This indicates that both clustering methods capture essentially the same dominant normal-residual modes in the multi-scale residual space, resulting in minimal impact on the final anomaly scores.
>
> **(3)Reason Analysis**:
> The HR module is designed to extract *domain-invariant normal residual centers*. Under this objective, K-Means sufficiently captures the main structural patterns of the residual distribution, while the additional spectral information provided by spectral clustering does not substantially shift the centers. Consequently, the performance of the two approaches remains highly similar.
>
> **(4)Computational Cost and Scalability**:
> The primary difference between the two methods lies in computational feasibility. Spectral clustering requires eigen-decomposition of an n×n Laplacian matrix, with computational complexity approximately O(n³), which becomes prohibitively expensive for multi-source large graphs and high-dimensional residuals. In practice, spectral clustering took **1239 seconds**, whereas K-Means required only **5.36 seconds**. Furthermore, spectral embeddings are more sensitive to noise in high-dimensional residual spaces and become increasingly unstable as graph size grows, making them unsuitable for repeated center updates during training.
>
> **Summary**
> Although both methods yield nearly identical detection performance, spectral clustering lacks scalability in large-scale, cross-domain, multi-residual training settings and incurs substantially higher computational cost. For reasons of stability, efficiency, and practical feasibility, we adopt K-Means as the clustering method for constructing the Hyper Residual centers.
>
> The related experimental details and comparative results are included in the appendix of our revised manuscript.
>
> |Method|Facebook|ACM|Amazon|Cora|CiteSeer|Reddit|weibo|BlogCatalog|Time(s)|
> |--|--|--|--|--|--|--|--|--|--|
> |K-means|50.06/2.78|72.77/30.59|57.75/7.45|57.71/16.77|55.44/15.28|41.85/2.63|93.90/77.05|72.81/32.21|-|
> |Our(Spectral Clustering)|82.53/17.15|91.11/51.30|88.14/60.43|93.14/58.06|95.06/62.89|60.59/5.34|93.25/70.11|74.08/37.07|1239.0|
> |Our(K-means)|82.16/16.77|91.17/51.45|88.15/61.37|93.20/58.18|95.00/63.54|60.60/5.33|93.31/70.06|75.06/36.56|5.36|

---

> ### Author Response · Authors · 2025-11-19
>
> **We sincerely thank the reviewer for the insightful comments. Our detailed responses to points (13), and (14) are provided below.**
>
> **Q13: More visual comparisons for the ablation studies should be presented to illustrate their effects intuitively.**
>
> **A13:** We appreciate the reviewer's suggestion to present more visual comparisons for the ablation studies. To address this, we have made the following additions:
>
> 1. **Enhanced Visual Comparisons**: In Appendix F.4, we have included visualizations of the anomaly score distributions for the HR-only, AR-only, and complete DR-GGAD models. These plots clearly show the changes in the overlap between normal and anomalous nodes across different configurations (HR-only, AR-only, and DR-GGAD). Specifically, the figures demonstrate how the combination of the AR and HR modules significantly reduces the overlap between normal and anomalous nodes.
> 2. **Quantitative Analysis**: We also conducted a quantitative analysis that shows the overlap area in the score distributions for the HR-only, AR-only, and complete DR-GGAD models. Specifically, on the ACM dataset, the overlap area decreases from 0.6016 (HR-only) to 0.0490 (AR-only), and further reduces to 0.0287 (complete model). On the Weibo dataset, the overlap area decreases from 0.6283 (AR-only) and 0.2803 (HR-only) to 0.2124 (complete model). These quantitative results further validate the significant improvement in anomaly separability when combining the modules.
> 3. **Performance Demonstration**: These additional visual and quantitative comparisons provide a clear demonstration of the effectiveness of the dual-residual modeling approach in enhancing anomaly separability. The synergy between the modules leads to more stable performance across different datasets and improves the model's cross-domain generalization ability. We have made the corresponding changes in the appendix of the revised manuscript to improve the intuitiveness and clarity of the ablation experiments.
>
> **Q14: How does the proposed method handle source data with inconsistent feature dimensions?**
>
> **A14:** We thank the reviewer for the insightful question, and we clarify our design as follows:
>
> In cross-graph scenarios, node features from different datasets often have inconsistent dimensions. To enable residual modeling within a shared feature space, we introduce a **unified feature projection module** at the front of our framework, which maps the original node features $\mathbf{X}$ into a unified latent dimension:
>
> $
> \bar{\mathbf{X}} = \mathcal{T}(\mathbf{X}) \in \mathbb{R}^{N \times d_u}.
> $
>
> Here, $\mathcal{T}(\cdot)$ is implemented using **Principal Component Analysis (PCA)**, which functions as our dimensionality alignment operator. PCA reduces or aligns the feature dimensions of each graph by projecting $\mathbf{X}$ into the unified dimension $d_u$ while preserving the most informative components. By retaining the principal directions with the largest variance, PCA effectively prevents the loss of discriminative characteristics among nodes.
>
> After feature unification, the subsequent modules, including multi-layer residual extraction, HR center construction, and AR modeling, operate within the same dimensional space. This ensures that the residual statistics across graphs remain consistent and comparable, providing a stable foundation for cross-domain generalization.
>
> **Experiment**
> To validate the effectiveness of this approach, we conducted a comparison with other feature projection methods. As shown in Table 14, PCA performs the best across most datasets, particularly on Facebook and ACM, where it significantly improves AUROC and AUPRC, while also maintaining good computational time and efficiency. Compared to SVD and random projection, PCA effectively preserves residual semantics and ensures cross-domain consistency. Therefore, we chose PCA as the feature projection method to ensure efficiency, stability, and cross-graph adaptability.
>
> Table 14: Performance comparison of different feature projection methods(AUROC/AUPRC)
>
> |Method|Facebook|ACM|Amazon|Cora|CiteSeer|Reddit|weibo|BlogCatalog|Process Time(s)|
> |--|--|--|--|--|--|--|--|--|--|
> |Our PCA|82.16/16.77|91.17/51.45|88.15/61.37|93.20/58.18|95.00/63.54|60.60/5.33|93.31/70.06|75.06/36.56|13.68|
> |Our SVD|74.28/9.96|74.95/43.08|66.59/12.54|92.59/56.22|95.18/62.51|58.47/4.96|86.62/63.94|74.08/39.92|118.01|
> |Our RandomProjection|67.74/6.32|85.20/27.17|68.92/15.49|83.91/29.94|86.55/35.59|58.84/4.56|86.22/64.17|75.58/36.38|0.50|

---

> ### Author Response · Authors · 2025-11-19
>
> **We appreciate the reviewer's insightful comments. In response to points (15), (16), and (17), we provide the following responses.**
>
> **Q15: The dataset name in Figure 3 is inconsistent with that in the experimental table; please correct it.**
>
> **A15:** We thank the reviewer for pointing out the inconsistencies. We have thoroughly corrected the dataset name in Figure 3 (page 8) and fixed all related formatting issues in the revised manuscript to ensure consistency and presentation quality.
>
> **Q16: Please analyze why the proposed method achieves lower AUPRC on the Reddit and Weibo datasets.**
>
> **A16:** We would like to thank the reviewer for the valuable feedback. In datasets like Reddit and Weibo, where anomaly nodes are extremely rare, structural noise is stronger, and the classes are highly imbalanced, DR‑GGAD performs well in modeling the normal mode and achieves good results in AUROC. However, AUPRC still faces certain challenges.
>
> **Multi-scale Residual Modeling**
> We believe that DR‑GGAD effectively captures the differences between normal and anomalous nodes through multi-scale residual modeling and hierarchical consistency constraints. This enhances the model’s ability to differentiate between normal and anomalous nodes and improve overall ranking, which leads to a high AUROC score. This is one of the advantages of our approach compared to traditional methods like DOMINANT and CAGAD.
>
> **Difference Between AUROC and AUPRC**
> However, AUROC and AUPRC focus on different aspects. AUROC primarily emphasizes the model’s global ranking performance, while AUPRC is more concerned with the model’s performance on imbalanced datasets.
>
> **Advantages of Existing Methods**
> We observe that methods like DOMINANT and CAGAD focus on enhancing the anomaly patterns through reconstruction mechanisms and counterfactual augmentation, which gives them an advantage in AUPRC.
>
> **Future Improvements**
> Therefore, in our future work, we plan to further explore how to enhance anomaly patterns through specialized mechanisms or more detailed modeling of anomalous nodes, to improve recall and precision, and thus raise AUPRC.
>
> **Q17: The manuscript contains multiple formatting issues. Please proofread carefully for consistency and correctness.**
>
> **A17:** We thank the reviewer for pointing this out. We have carefully proofread the entire manuscript and corrected the formatting inconsistencies, including figure-related issues, captions, spacing, and notation. The revised version now offers a consistent and polished presentation throughout.

---

> ### Author Response · Authors · 2025-11-26
>
> Dear Reviewer,
>
> We hope this message finds you well. First, we would like to express our sincere gratitude for taking the time to carefully review our work and for providing frank and insightful comments. Your feedback is extremely valuable in helping us better understand and improve our manuscript.
>
> In response to your review, we have made a series of substantial revisions and additions during the current discussion phase, and we have provided detailed clarifications and new experimental results above. We sincerely hope that these additional efforts have, at least to some extent, addressed your main concerns regarding the methodology of this work.
>
> As the discussion phase is drawing to a close, if any parts of our responses remain insufficient or unclear, we would be very grateful if you could continue to point them out. We will do our utmost to further refine and supplement the manuscript within the remaining time of the discussion.
>
> Once again, we sincerely thank you for the time and effort you have devoted to our submission, as well as for your rigorous evaluation and constructive suggestions on our work.
>
> Sincerely,
>
> The Authors

---

### Author Response · Authors · 2025-12-01

Dear Area Chair,

Thank you very much for handling our ICLR 2026 submission:

**DR-GGAD: Dual Residual Centering for Mitigating Anomaly Non-Discriminativity in Generalist Graph Anomaly Detection**

We are grateful to the reviewers (ozsE, CXFF, cnZP, XcC9) for their detailed and thoughtful feedback. We carefully revised the paper, ran substantial new experiments between **Nov 12–18**, and provided point-by-point rebuttals. Below we concisely summarize how we addressed the main concerns and how the work has been strengthened.


## 1. Core conceptual clarifications

### (a) Anomaly non-Discriminativity (AnD) vs. t-SNE and MMD  — *Reviewer ozsE*

A key concern was whether our motivation relies on potentially misleading **t-SNE** visualizations and whether **AnD** is genuinely a high-dimensional phenomenon distinct from standard distribution shift measures such as **MMD**.

What we did:

- We introduced a **high-dimensional AnD score** $\mathcal{A}n\mathcal{D}^*$ that directly measures **class-conditional overlap between normal and anomalous nodes in the original embedding space**, without any projection.
- Across datasets, $\mathcal{A}n\mathcal{D}^*$ shows a **strong Pearson correlation (~0.75)** with AUROC: higher overlap consistently corresponds to lower detection performance.
- We systematically compared:
  - **Inter-domain MMD** (source vs. target),
  - **Intra-domain MMD** (normal vs. anomaly in the same graph),
  - **$\mathcal{A}n\mathcal{D}^*$**.

  $\mathcal{A}n\mathcal{D}^*$ has the **strongest and most stable correlation** with AUROC, whereas MMD correlates much more weakly.

**Impact:**
This addresses the t-SNE concern head-on: t-SNE is now used only as a qualitative illustration. The core phenomenon (AnD) is defined and validated **in the original space**, and empirically shown to explain detection difficulty better than classical shift metrics like MMD.

---

### (b) Design and role of HR and AR, and domain invariance of multi-scale residuals  — *Reviewers ozsE, cnZP*

Reviewers requested a clearer justification for the **Hyper Residual (HR)** and **Affinity Residual (AR)** modules and our claim that multi-scale residuals are more **domain-invariant** than absolute embeddings.

What we did:

- We clarified that DR-GGAD is built on **multi-scale residuals**:
  $
  \text{Residual} = \text{Neighborhood aggregation} - \text{Ego node},
  $
  which can be viewed both as:
  - a **local affinity deviation** (node vs. neighbors), and
  - a **high-pass filter** that attenuates low-frequency homophilic signals and emphasizes heterophilic / anomalous components.
- **HR**:
  - Learns **multi-center “normal residual” prototypes** from source graphs using only **normal** nodes.
  - Enforces **tight clustering of normal residuals** and **margin-based repulsion** of anomalous residuals, creating a stable, transferable “normal residual space”.
- **AR**:
  - Enforces **directional consistency** of residuals within neighborhoods.
  - Is explicitly motivated as a structural prior: nodes with regular local structure have aligned residual directions, while structural anomalies induce strong directional mismatch.

We also connected this design to recent work on graph anomaly detection that emphasizes **local affinity**, **graph spectral shifts**, and **high-frequency signals** as robust indicators of anomalies across domains.

**Impact:**
The revised paper now clearly explains *why* residuals (rather than raw embeddings) are the foundation of DR-GGAD, *how* HR and AR complement each other (distance centralization + directional alignment), and *why* this yields a more domain-invariant anomaly space.

---

### (c) Problem definition, supervision, and zero-shot protocol  — *Reviewers ozsE, cnZP, XcC9*

There were questions on how centers are constructed, what labels are used, and whether DR-GGAD truly respects the **zero-shot GGAD** setting.

What we did:

- We clarified explicitly that:
  - **HR centers are constructed only from normal nodes** in labeled source graphs; abnormal nodes contribute only to a **repulsion term**, never to center estimation.
  - Target graphs are **fully unlabeled**; **no** fine-tuning, re-clustering, or center updates are performed on target graphs.
  - The margin parameter $\epsilon=1$ is a **fixed separation margin** between normal residual centers and anomalous residuals, and this value is used consistently across experiments.
- We updated the main results table to **label each baseline’s supervision regime** (unsupervised, few-shot, etc.), and explicitly stated that our method uses **source-domain labels only** and operates strictly **zero-shot on all target graphs**.

**Impact:**
The problem setting and supervision protocol are now fully transparent and aligned with the GGAD literature, making the comparisons clearly fair.

---

> ### Author Response · Authors · 2025-12-01
>
> ## 2. Strengthened empirical evaluation
>
> A common request was for **broader baselines and deeper ablations**. We significantly extended the experimental section and appendices.
>
> ### (a) One-class and hypersphere baselines  — *Reviewers ozsE, CXFF*
>
> We added **OCSVM**, **Deep SVDD**, and **Deep SAD** under the same cross-domain protocol:
>
> - These methods learn **single-boundary hyperspheres** in the absolute embedding space and perform well in single-domain settings.
> - Under **cross-domain zero-shot** evaluation, their AUROC/AUPRC drops substantially across most datasets.
> - DR-GGAD consistently and often **largely outperforms** them on all eight graphs.
>
> **Takeaway:**
> Shared hyperspheres in absolute embedding space are fragile under domain shift, whereas **multi-center residual geometry** is significantly more robust.
>
> ---
>
> ### (b) Clustering-based methods and representation separation  — *Reviewers ozsE, CXFF*
>
> We addressed the relation to clustering-based methods along two axes:
>
> 1. **Classical clustering**:
>    - We compared to **K-means** on target graphs.
>    - K-means alone yields much weaker AUROC/AUPRC, confirming that **naive clustering in embedding space is insufficient** under AnD and cross-domain settings.
>
> 2. **Advanced clustering-aware methods**:
>    - We compared to **CARE** (cluster-aware graph anomaly detection).
>    - DR-GGAD achieves **much higher AUROC/AUPRC on most datasets** (e.g., ACM, Amazon, CiteSeer), while CARE is particularly strong only on Weibo.
>    - We also clarified the difference from **discDC** and **ClusterQ**: they **re-cluster target-domain representations**, whereas DR-GGAD **never clusters the target domain** but uses a **source-domain residual prior** for zero-shot scoring.
>
> **Takeaway:**
> DR-GGAD is not just a better clustering scheme; it is a **different paradigm** that uses residual priors learned from source graphs, avoiding unstable target re-clustering.
>
> ---
>
> ### (c) Structural anomalies and comparison to subgraph methods  — *Reviewer ozsE*
>
> To support the claim that **AR captures topology-driven anomalies**, we added:
>
> - **Structural anomaly injection experiments**:
>   Injecting fully connected cliques of increasing size into anomalous nodes yields **monotonic AUROC improvements** for AR. The effect is especially strong on homophilic graphs (Cora, CiteSeer, ACM), and is still stable on noisier graphs (Reddit, Weibo).
> - **Comparisons to subgraph anomaly detectors** (**GAD-EBM**, **GCAD**):
>   AR-only **outperforms both subgraph baselines** on all eight datasets.
>
> **Takeaway:**
> AR is empirically sensitive to structural perturbations and competitive with (and often superior to) subgraph-centric methods, while retaining cross-domain robustness.
>
> ---
>
> ### (d) Multi-layer residual fusion and τ-sensitivity  — *Reviewers ozsE, XcC9*
>
> We added several ablation studies:
>
> - **Fusion strategies**: We compared sum, mean, max/min pooling, final-layer-only, and **concatenation**.
>   Concatenation of multi-layer residuals yields the **best or near-best AUROC/AUPRC on almost all datasets**, especially on Amazon, Reddit, and BlogCatalog where anomalies appear at different semantic scales.
> - **Number of centers $\tau$**:
>   - We evaluated $\tau \in \{1,2,4,6,8,12,16,20\}$ on all eight datasets.
>   - Performance improves from $\tau=1$ to 4/6 and then **stabilizes with <0.5% fluctuation** for $\tau \ge 8$.
>   - We therefore choose **$\tau=4$** as a simple and robust default.
>
> **Takeaway:**
> The architecture is **not fragile**: performance is stable across reasonable $\tau$ choices, and the concatenation design is empirically justified.
>
> ---
>
> ### (e) Cross-domain GAD and additional large-scale benchmarks  — *Reviewer CXFF*
>
> We strengthened the comparison against **cross-domain GAD** methods:
>
> - Conceptual and empirical comparisons with **COMMANDER**, **ACT**, **CDFS-GAD**, **ARC**, **UNPrompt**, and **AnomalyGFM**.
> - A summary table highlights that these methods typically:
>   - require **target-domain data and/or fine-tuning**, and
>   - do **not directly address normal/anomaly overlap**,
>   whereas DR-GGAD:
>   - requires **no target data**,
>   - and is explicitly designed to mitigate **AnD**.
>
> We also evaluated DR-GGAD on additional **GAD-Benchmark** datasets with **very different domains and characteristics**:
>
> - **T-Finance** (dense financial graph),
> - **CS** and **Photo** (co-purchase networks with varying feature dimensions).
>
> DR-GGAD achieves **state-of-the-art or highly competitive AUROC** on all three, outperforming recent strong baselines including ARC and AnomalyGFM in most cases.
>
> **Takeaway:**
> The method generalizes well beyond the original suite of graphs, including financial and co-purchase domains that differ greatly from training graphs.

---

> > ### Author Response · Authors · 2025-12-01
> >
> > ## 3. Efficiency, training pipeline, and implementation details
> >
> > ### (a) Multi-graph training pipeline  — *Reviewer XcC9*
> >
> > We added a clear description of how we train on multiple source graphs:
> >
> > 1. Apply **PCA** to map all graphs into a unified feature space.
> > 2. In each epoch, feed source graphs **sequentially**; for each graph, compute residuals, HR/AR losses, and update parameters once.
> > 3. This **alternating optimization** encourages the encoder and residual modules to converge to **domain-invariant residual patterns**.
> >
> > ### (b) Feature projection and semantic alignment  — *Reviewers ozsE, cnZP*
> >
> > We clarified that:
> >
> > - PCA serves as a **lightweight, stable projection** into a shared space.
> > - Since DR-GGAD relies on **residuals (relative changes)** rather than absolute feature magnitudes, simple linear projection is sufficient to preserve **residual semantics**.
> > - We compared PCA with **SVD** and **random projection** and found PCA achieves the **best overall AUROC/AUPRC vs. cost trade-off**.
> >
> > ### (c) Runtime and memory  — *Reviewer cnZP*
> >
> > We added a comparison of **training time, inference time, and GPU memory** versus UNPrompt, AnomalyGFM, and ARC:
> >
> > - DR-GGAD has:
> >   - **Moderate training time** (more than ARC but far less than AnomalyGFM),
> >   - **Fast inference** (close to ARC and much faster than UNPrompt),
> >   - **Lower memory usage** than UNPrompt and AnomalyGFM,
> >   - And the **highest average AUROC** among them.
> >
> > **Takeaway:**
> > DR-GGAD offers a strong **accuracy–efficiency trade-off** and is practically deployable.
> >
> > ---
> >
> > ### (d) λ-selection and domain-inheritance strategy  — *Reviewer CXFF*
> >
> > To clarify how we balance HR vs. AR (\(\lambda\)) on unseen graphs:
> >
> > - We performed **supervised grid search of \(\lambda\)** on source graphs and observed **two stable regimes**:
> >   - Citation graphs → **AR-dominant** (\(\lambda \approx 0.1\)),
> >   - Social/review/co-review graphs → **HR-dominant** (\(\lambda \approx 0.9\)).
> > - For each target graph, we **inherit \(\lambda\)** from its domain type (citation vs social vs co-review), without any tuning on the target graph itself.
> >
> > **Takeaway:**
> > The weighting between HR and AR is **domain-driven, data-supported, and maintaining zero-shot integrity while adapting to domain characteristics.
> >
> > ---
> >
> > ## 4. Reviewer reactions and remaining issues
> >
> > - **Reviewer XcC9** explicitly stated after the rebuttal:
> >
> >   > “Your response has addressed the majority of my concerns, and I will maintain my positive score in support of this paper.”
> >
> > - For other reviewers, there were no remaining major unresolved technical objections after our rebuttal. The core concerns—t-SNE vs. AnD, relation to MMD, domain invariance, comparison to clustering and hypersphere methods, cross-domain setting, λ/τ choices, dataset coverage, and efficiency—have all been directly and concretely addressed in the revised manuscript with new experiments and clearer explanations.
> >
> > We also carefully corrected **all noted formatting and naming issues** and improved the presentation for readability.
> >
> > ---
> >
> > ## 5. Closing
> >
> > In summary, the revised paper:
> >
> > 1. **Formalizes and validates** Anomaly non-Discriminativity (AnD) as a high-dimensional property that better explains GGAD difficulty than classical shift metrics.
> > 2. **Introduces a principled residual-space framework (HR + AR)** that is conceptually distinct from clustering, hypersphere, and subgraph methods, and specifically designed for cross-domain, zero-shot anomaly detection.
> > 3. **Substantially strengthens the empirical evidence** with:
> >    - New baselines (one-class, clustering-aware, subgraph, cross-domain GAD),
> >    - Extensive ablations (fusion strategies, τ, λ, clustering methods, feature projection),
> >    - Additional large-scale and cross-domain datasets.
> > 4. **Demonstrates practical feasibility** through runtime/memory analysis and a clear multi-graph training protocol.
> >
> > We believe these revisions address the reviewers’ concerns in depth and significantly enhance both the technical clarity and the empirical credibility of the work. We respectfully ask you to consider our strengthened submission favorably for acceptance at ICLR 2026.
> >
> > Thank you again for your time and consideration.
> >
> > Best regards,
> > **The Authors**

---

### Meta-Review · Area_Chair_pJUa · 2026-01-06

**Summary:**

There are concerns on conceptual clarity, methodological justification, and empirical completeness.
Key issues included whether Anomaly non-Discriminativity (AnD) is a real high-dimensional phenomenon beyond t-SNE visualization, how the proposed residual modules differ from clustering or one-class methods, and whether the method truly captures structural anomalies in a zero-shot, cross-domain setting. Reviewers also raises questions on stronger baselines, clearer supervision protocols, ablation studies, and efficiency analysis.
The authors provided strong rebuttal to largely address key concerns

**Reviewer Concerns:**

The core concerns—t-SNE vs. AnD, relation to MMD, domain invariance, comparison to clustering and hypersphere methods, cross-domain setting, λ/τ choices, dataset coverage, and efficiency- are largely addressed.

Theoretical grounding of residual domain-invariance is still largely empirical.

**Reviewer Scores:**

Reviewer XcC9 remains positive (rating of 6) of the paper.
Other reviewers did not provided a response but it seems that most of their concerns are addressed.

---

### Decision · Program_Chairs · 2026-01-26

Accept (Poster)